# Hexosamine biosynthetic pathway and O-GlcNAc-processing enzymes regulate daily rhythms in protein O-GlcNAcylation

Xianhui Liu[1], Ivana Blaženović[2], Adam J. Contreras[1], Thu M. Pham[1], Christine A. Tabuloc[1], Ying H. Li[1], Jian Ji[3], Oliver Fiehn [2] & Joanna C. Chiu [1✉]

The integration of circadian and metabolic signals is essential for maintaining robust circadian rhythms and ensuring efficient metabolism and energy use. Using *Drosophila* as an animal model, we show that cellular protein O-GlcNAcylation exhibits robust 24-hour rhythm and represents a key post-translational mechanism that regulates circadian physiology. We observe strong correlation between protein O-GlcNAcylation rhythms and clock-controlled feeding-fasting cycles, suggesting that O-GlcNAcylation rhythms are primarily driven by nutrient input. Interestingly, daily O-GlcNAcylation rhythms are severely dampened when we subject flies to time-restricted feeding at unnatural feeding time. This suggests the presence of clock-regulated buffering mechanisms that prevent excessive O-GlcNAcylation at non-optimal times of the day-night cycle. We show that this buffering mechanism is mediated by the expression and activity of GFAT, OGT, and OGA, which are regulated through integration of circadian and metabolic signals. Finally, we generate a mathematical model to describe the key factors that regulate daily O-GlcNAcylation rhythm.

[1] Department of Entomology and Nematology, College of Agricultural and Environmental Sciences, University of California, Davis, CA, USA. [2] West Coast Metabolomics Center, University of California, Davis, CA, USA. [3] School of Food Science, State Key Laboratory of Food Science and Technology, National Engineering Research Center for Functional Foods, School of Food Science Synergetic Innovation Center of Food Safety and Nutrition, Jiangnan University, Wuxi, Jiangsu, China. ✉email: jcchiu@ucdavis.edu

Circadian clocks exist in organisms from all domains of life. These endogenous timers perceive daily rhythms in environmental and nutrient signals and control the timing of physiological and metabolic processes to optimize energy homeostasis[1–3]. At the behavioral level, feeding-fasting cycles exhibit robust daily rhythms that are regulated by the circadian clock[4,5]. At the molecular level, circadian clocks regulate the rhythmicity of cellular metabolic processes to anticipate feeding-induced nutrient influx to ensure efficient metabolism and energy use[1,6]. The circadian clock has been shown to drive daily rhythmic expression of metabolic genes involved in glycolysis, pentose phosphate pathway, gluconeogenesis, lipid oxidation and storage[1,7]. Once metabolized, macronutrients such as sugars, amino acids and lipids, or the lack of nutrients during fasting period can in turn regulate appropriate nutrient-sensing signaling pathways, such as those regulated by insulin/target of rapamycin (TOR), adenosine-monophosphate-activated protein kinase (AMPK), glucagon, adipokines, and autophagy, to orchestrate downstream physiological functions[1,6,8]. The coordinated interplay between cellular processes that are regulated by the circadian clock and those regulated primarily by metabolic signaling is essential for maintaining the robustness of circadian physiology and organismal health[9,10].

Rhythmic nutrient signals from clock-controlled feeding activity not only feedback to maintain the robustness of the circadian oscillator through metabolic regulation, but can also feedforward to regulate rhythmicity of cellular processes beyond the circadian oscillator[8,9]. There are many examples illustrating metabolic regulation of key components of the circadian oscillator[1,11–13]. For example, AMPK promotes the degradation of CRYPTOCHROME 1 (CRY1), a key component of the mammalian clock[12], while in *Drosophila*, GLYCOGEN SYNTHASE KINASE 3β (GSK3β) facilitates nuclear entry of the PERIOD-TIMELESS (PER-TIM) transcriptional repressor complex to regulate circadian period length and output[11]. Beyond the circadian oscillator, daily oscillation of reactive metabolites can impose timely regulation on transcriptional and chromatin dynamics. Metabolism modulates the global chromatin landscape by rhythmically providing essential substrates, such as acetyl-CoA for acetylation and S-adenosyl methionine for methylation[14,15]. Furthermore, metabolites can serve as cofactors to directly affect activities of chromatin remodelers and histone modifiers to regulate the circadian transcriptome. Sirtuins (SIRT), a class of proteins with NAD$^+$-activated deacetylase activities, have been identified as key integrators of circadian and metabolic signaling due to their roles in regulating key circadian clock proteins and the chromatin landscape[13].

Despite significant progress in understanding the coordination between the circadian clock and metabolism in regulating circadian physiology, our knowledge is far from complete. Specifically, emerging data showing that protein O-linked N-Acetylglucosaminylation (O-GlcNAcylation), a nutrient-sensitive post-translational modification (PTM), is highly prevalent and can regulate diverse cellular processes[16–19], prompted us to hypothesize that rhythmic nutrient influx through clock-controlled feeding-fasting cycles may regulate time-of-day-specific O-GlcNAcylation and cellular protein functions. Uridine diphosphate-N-acetyglucosamine (UDP-GlcNAc) is the end-product of the hexosamine biosynthetic pathway (HBP) and the donor substrate that enables modifications at serine and threonine residues with O-GlcNAcylation. HBP is considered the most generalized sensor for metabolic status as it integrates metabolites from breakdown of glucose, amino acids, lipids, and nucleic acids[19]. Together with the fact that there is extensive interplay between O-GlcNAcylation and phosphorylation to regulate cellular protein functions[16,17] and recent findings that a significant portion of the phosphoproteomes in mice and flies exhibit daily oscillations[20–22], we sought to investigate if protein O-GlcNAcylation represents an important post-translational mechanism that integrates circadian and metabolic regulation to coordinate circadian physiology.

In previous studies, we and others have shown that a number of key clock transcription factors in flies and mice exhibit daily O-GlcNAcylation rhythms that regulate their subcellular localization, activity, and stability[23–26]. However, it is not clear whether cellular proteins beyond the circadian oscillator also exhibit daily O-GlcNAcylation rhythms. In this study, we observed that the O-GlcNAcylation of total nuclear proteins oscillates over a 24-h period in wild type (WT) *Drosophila*. Nuclear protein O-GlcNAcylation rhythms showed strong correlation with rhythms in food intake and HBP metabolites, suggesting that they could be driving protein O-GlcNAcylation rhythms. We then manipulated timing of nutrient input using time-restricted feeding (TRF) to establish causal relationships between feeding time, HBP metabolites, and protein O-GlcNAcylation rhythm. Despite the importance of feeding time, we observed that the phase of protein O-GlcNAcylation rhythms did not simply shift when feeding occurs at unnatural time window. Rather, the amplitude of the daily O-GlcNAcylation rhythm dampens significantly. This suggests that clock-controlled buffering mechanisms exist to limit excessive O-GlcNAcylation if animals feed during non-optimal time of the day-night cycle. By characterizing the daily activity of HBP, we found that glutamine-fructose-6-phosphate amidotransferase (GFAT), the rate-limiting enzyme of HBP, is regulated by multiple clock-controlled mechanisms to influence HBP output and protein O-GlcNAcylation rhythm. We observed that *gfat2* mRNA is induced by clock-controlled food intake and GFAT enzyme activity is regulated at the post-transcriptional level, likely by time-of-day-specific phosphorylation. Furthermore, we found that O-GlcNAc transferase (OGT) and O-GlcNAcase (OGA), the two key O-GlcNAc-processing enzymes, are also regulated by circadian and metabolic signals at the transcriptional and protein levels. In summary, our results provide insights into the role of HBP and O-GlcNAc-processing enzymes in integrating circadian and metabolic signals to regulate daily rhythms in cellular protein O-GlcNAcylation. Our results shed light on the health benefits of TRF and deleterious effects of non-optimal meal times, which are common in modern society.

## Results

**O-GlcNAcylation of nuclear proteins exhibits daily rhythmicity.** A number of key circadian clock proteins have previously been observed to exhibit daily rhythms in O-GlcNAcylation that modulate their time-of-day-specific functions[23–26]. We hypothesized that this phenomenon may be more widespread and could serve as an important mechanism that regulates daily rhythms in protein structure and function. Using chemoenzymatic labeling[27], a strategy we have previously employed to examine PERIOD (PER) O-GlcNAcylation[26], we conjugated biotin tags to O-GlcNAc groups on nuclear proteins extracted from wild type (WT; $w^{1118}$) male fly bodies. We observed that a collection of proteins, ranging from 37 to 250 kD, are O-GlcNAcylated in the nuclei extracted from body tissues (Fig. 1a, top panel). The band detected in unlabeled samples (Fig. 1a, middle panel) was deemed to be non-specific and excluded during quantification (Fig. 1a, b) (Please refer to the "Methods" section for the use of unlabeled samples to identify non-specific signal). Our results showed that O-GlcNAcylation of nuclear proteins exhibited a robust daily rhythm, suggesting that timely metabolic input may regulate the function of cellular proteins through this post-translational modification. O-GlcNAcylation analyses were conducted only

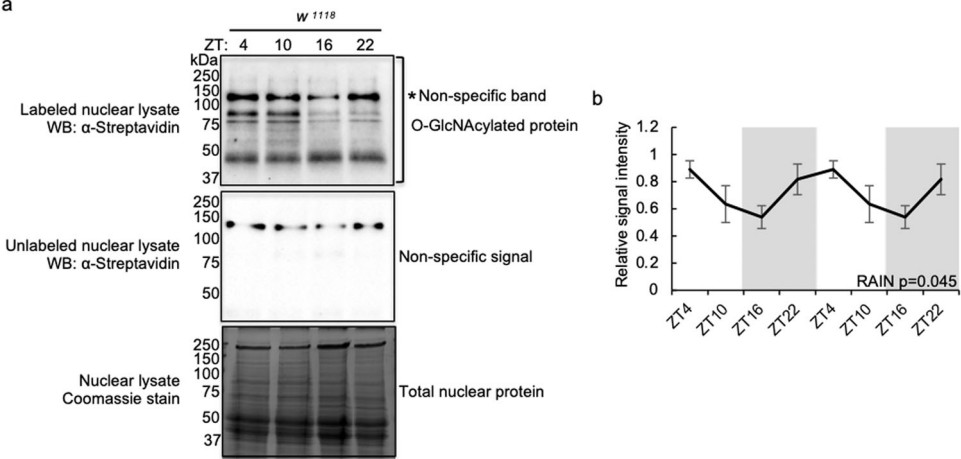

**Fig. 1 Protein O-GlcNAcylation oscillates over a 24-h day-night cycle under ad libtum condition. a** Western blots showing daily rhythms in O-GlcNAcylation of nuclear proteins extracted from body tissues of wild type ($w^{1118}$) flies fed *ad libtum*. O-GlcNAcylation is detected using chemoenzymatic labeling in combination with Western blotting using α-streptavidin. Unlabeled samples (middle panel) were processed in parallel to labeled samples (top panel) to identify non-specific signal. Total nuclear proteins were stained using Coomassie blue (bottom panel) and were used for normalization. The asterisk (top panel) denotes non-specific signal. Each biological replication was performed independently with similar results ($n = 4$). **b** Quantification of nuclear protein O-GlcNAcylation in (**a**). Data are double plotted and presented as mean ± SEM ($n = 4$; $p = 0.045$; RAIN). α-streptavidin signal of the whole lane was quantified, and the non-specific signal was used for background deduction prior to normalization. Data are double plotted and presented as mean ± SEM. The grey shading indicates the dark phase of each 12 h light:12 h dark (LD) cycle. ZT Zeitgeber time (hr). ZT0 denotes lights-on time.

on body tissues as we expected them to be more metabolically sensitive[4,28,29]. In addition, since the labeling method we used cannot discriminate between GlcNAc groups on N-linked and O-linked glycoproteins, only nuclear proteins were analyzed to avoid contamination from N-glycans in the endoplasmic reticulum, the Golgi apparatus and on the cell surface[30]. Nevertheless to ensure there were no contaminants of N-Glycans, we compared the level of biotin conjugation in nuclear lysates with or without PNGase F treatment, since PNGase F specifically cleaves N-glycans from proteins. We observed that nuclear lysates with or without PNGase F treatment showed similar level of chemoenzymatic biotin labeling (Supplementary Fig. 1a, b).

**Daily rhythmicity of HBP metabolites correlates with feeding rhythm.** Next we investigated whether feeding-fasting rhythm drives daily oscillation of nutrient input and impacts the timing of cellular protein O-GlcNAcylation. We first measured timing of daily feeding activities of WT flies using the CApillary FEeder (CAFE) assay[4,31]. Male and female flies were provided with food *ad libitum* and monitored separately over two day-night cycles. Although both sexes exhibited robust feeding-fasting rhythm, peak feeding activity was observed around ZT24 in males as previously reported[4] (Fig. 2a) while females exhibited a phase delay relative to males with a peak around ZT12 (Supplementary Fig. 2a). Because of the observed sexual dimorphism in the timing of feeding activity, only male flies were used in subsequent experiments.

We harvested fly tissues over a 24-h period at 6-h intervals and screened primary metabolites by GC-TOF MS on head and body tissues separately to determine if there is correlation between daily rhythms in feeding and metabolites, especially those in the HBP. We detected 573 metabolites in total, with 167 known and 406 unknown compounds (Supplementary Data 1). Replicates that exhibited 10-fold differences of total peak intensity in comparison to other replicates were excluded in the analysis (Supplementary Fig. 2b, c). 93.0% of the total compounds cycle in fly bodies while only 12.7% cycle in heads (RAIN $p < 0.05$, Supplementary Data 2) (Fig. 2b). This difference in rhythmicity between head vs body metabolites is likely due to the blood-brain

barrier. Similar to the previous circadian metabolomic study conducted in flies[29], most of the cycling metabolites peak around ZT10, which is subsequent to peak time of food intake, corroborating an anticipated relationship between feeding and nutrient influx (Fig. 2c and Supplementary Data 1). The majority of known compounds, including carbohydrates, amino acids and a small number of lipids, belong to primary metabolic pathways. As we are especially interested in protein O-GlcNAcylation, we targeted the HBP (Fig. 2d) for more in-depth data analysis. HBP is a branch of glycolysis that produces UDP-GlcNAc, the key donor substrate for O-GlcNAcylation. We observed that the first four HBP metabolites and the end product, UDP-GlcNAc, displayed robust 24-h rhythmicity in body tissues (RAIN $p < 0.05$), while rhythmicity was weak or absent in heads (Fig. 2e, RAIN $p > 0.25$). The oscillation of UDP-GlcNAc, similar to the majority of the detected compounds, is strongly correlated to fly feeding rhythm, with maximum level occurring soon after peak food intake and the trough during the fasting period (Fig. 2f and Supplementary Fig. 3a, b, d and g). Furthermore, we observed strong correlations between the daily oscillations of UDP-GlcNAc vs. O-GlcNAcylation of nuclear proteins and between the timing of feeding activity vs. O-GlcNAcylation rhythm (Fig. 2g, h and Supplementary Figs. 3a–c, e, f, h and i). This supports our hypothesis that feeding-fasting cycle drives the oscillation of UDP-GlcNAc, which in turn promotes rhythmic protein O-GlcNAcylation. As our metabolomics data supports that body tissue is more metabolically sensitive and UDP-GlcNAc level oscillates robustly only in body tissue, we chose to perform all subsequent experiments only in fly body tissues.

**Feeding-fasting cycles regulate daily O-GlcNAcylation rhythm.** To establish causality between feeding-fasting cycles and rhythmic protein O-GlcNAcylation, we manipulated timing of food intake by time-restricted feeding (TRF) and monitored changes in O-GlcNAcylation patterns in fly body tissues. We divided WT flies into three feeding groups (Fig. 3a): (1) The *ad libitum* (AL) group had food available at all times; (2) the natural feeding (RF21-3) group was only fed between ZT 21 to ZT3, the natural fly feeding time according to our CAFE assay (Fig. 2a); (3) the

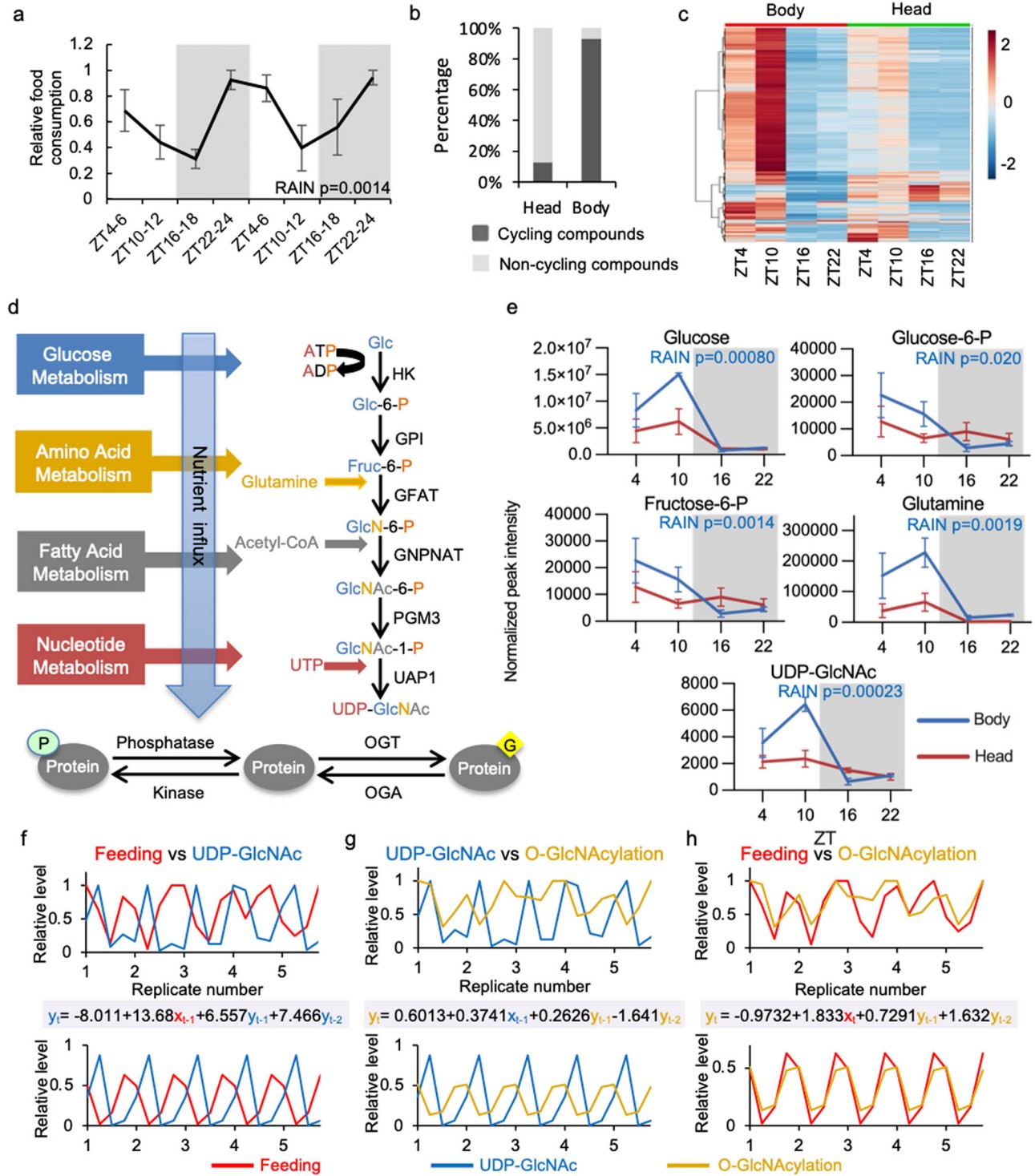

mistimed feeding group (RF9-15) was fed 12 h out of phase[28]. We expect that with a limited feeding window in the TRF groups, the amplitude of the O-GlcNAcylation rhythm may be higher in comparison to that in the AL group in which feeding is relatively less consolidated. We first used the CAFE assay to confirm that the feeding groups consumed similar amount of food daily (Fig. 3b). We then compared the feeding amount in TRF groups to that of the AL group during their respective feeding window to show that the TRF treatments indeed consolidated their daily food intake into the 6-h feeding window (Fig. 3c).

The daily O-GlcNAcylation profile of nuclear proteins in WT body tissue was determined using the chemoenzymatic labeling

method as described above. In the RF21-3 group, the oscillation of protein O-GlcNAcylation was strengthened, as indicated by comparing RAIN statistics of AL and RF21-3 groups (Fig. 3d top panel and e). The RF9-15 group, however, exhibited a severely dampened O-GlcNAcylation rhythm despite having access to food for the same duration of time and consuming roughly the same amount as the RF21-3 group (Fig. 3d top panel and e). These results derived from quantification of biotin-conjugated O-GlcNAcylated proteins detected using α-streptavidin were further confirmed by western blots using an α-O-GlcNAc antibody (Fig. 3d third panel from top and f). On one hand, results from TRF experiments support our hypothesis that feeding-fasting

**Fig. 2 Daily feeding-fasting cycle generates oscillation in metabolites in flies. a** Feeding rhythm of wild type ($w^{1118}$) male flies over two consecutive days as measured by CAFE assay. Data were normalized (peak feeding = 1) and presented as mean ± SEM ($n = 3$, 10 flies per biological replicate; $p = 0.0014$; RAIN). The grey shading indicates the dark phase of each day-night cycle. **b** Percentage of cycling and non-cycling metabolites in male fly heads and bodies detected by GC-MS ($n = 5$ or 6; $p < 0.05$; RAIN). **c** Heat map of all metabolites detected in male heads and bodies ($n = 5$ or 6). Normalized peak intensity of all metabolites were pareto scaled. Each line represents one detected compound. **d** Schematic showing hexosamine biosynthetic pathway (HBP) integrating cellular metabolic status to provide UDP-GlcNAc as output (format of diagram modified from Hart et al., 2011[16]). HK Hexokinase, GPI Phosphoglucose isomerase, GFAT Glutamine--fructose-6-phosphate aminotransferase, GNPNAT Glucosamine-phosphate N-acetyltransferase, PGM3 Phosphoacetylglucosamine mutase, UAP1 UDP-N-Acetyl glucosamine pyrophosphorylase 1, OGT O-GlcNAc transferase, OGA O-GlcNAcase. Glc glucose, Glc-6-P Glucose-6-phosphate, Fruc-6-P Fructose-6-phosphate, GlcN-6-P Glucosamine-6-phosphate, GlcNAc-6-P N-acetylglucosamine-6-phosphate, GlcNAc-1-P N-acetylglucosamine-1-phosphate, UTP Uridine triphosphate, UDP-GlcNAc Uridine diphosphate N-acetylglucosamine. **e** Line graphs showing daily oscillations of HBP metabolites in fly bodies and heads ($n = 5$ or 6). Data were presented as mean ± SEM. P values indicating rhythmicity (RAIN) of each of the five metabolites in fly bodies are presented on the upper right corner of each panel. $p > 0.25$ (RAIN) for all HBP metabolites in head samples. **f–h** Cross-correlation analysis between rhythms: (**f**) feeding and UDP-GlcNAc, (**g**) UDP-GlcNAc and O-GlcNAcylation, (**h**) feeding and O-GlcNAcylation. Top panels show the observed raw data with all biological replicates, while bottom panels indicate the rhythmic pattern extracted from the raw data using R. The correlation equations are shown between panels with the color of terms corresponding to each curve (For all the equations, $R^2 = 1$).

rhythm drives protein O-GlcNAcylation rhythm as consolidated feeding activity in the RF21-3 group led to more robust nuclear O-GlcNAcylation rhythm. On the other hand, since TRF at unnatural feeding time (RF 9-15) did not simply produce a phase shift of protein O-GlcNAcylation rhythm, we postulate that the circadian clock must play additional role(s) in regulating protein O-GlcNAcylation rhythm besides the regulation of feeding-fasting cycles.

**Circadian clock regulates the GFAT-catalyzed step in HBP.**
Combing through CirGRDB[32], a mammalian circadian transcriptomic database, and published *Drosophila* transcriptomics data (Supplementary Table 1), we found that all HBP enzymes have been identified as rhythmic genes in the mouse liver or fly head in at least one study, even though the phase of some transcripts may be variable between studies. Moreover, proteomic or phosphoproteomic analyses indicated that some of these HBP enzymes exhibit rhythmic protein expression or phosphorylation status[20,22,33]. We therefore reasoned that the circadian clock may regulate one or more of the HBP enzymes to modulate protein O-GlcNAcylation rhythm. To identify key HBP effector(s) of the circadian clock, we performed targeted metabolomic analysis using hydrophilic interaction liquid chromatography coupled to mass spectrometry (HILIC-MS/MS) on body tissues of WT flies subjected to TRF between ZT21-3 and ZT9-15. We reasoned that measurements of HBP metabolites in combination with TRF treatments would allow us to evaluate potential time-of-day differences in HBP enzyme activities and identify key clock-controlled mechanism(s) that facilitate rhythmic protein O-GlcNAcylation when food is consumed at natural feeding time. These mechanisms are also expected to limit excessive O-GlcNAcylation when food intake occurs during non-optimal time during the day-night cycle. Standard curves of HBP metabolites were generated to obtain more reliable concentration measurements for each metabolite (Supplementary Fig. 4a). We were able to detect 8 to 9 out of 13 metabolites in the HBP, but were not able to differentiate between GlcNAc-6-P and GlcNAc-1-P.

In both TRF groups, we observed robust cycling of all HBP metabolites (RAIN $p < 0.001$), except for glucosamine and glucosamine-6-phosphate. The cycling HBP metabolites were found to peak after the respective feeding time in both TRF groups, suggesting that metabolite levels are significantly elevated in response to nutrient influx (Fig. 4a). We explored whether there was a significant difference in daily rhythmicity between HBP metabolites in the two TRF groups and found that fructose-6-phosphate is the most statistically different metabolite between the two groups (Supplementary Data 3). In comparison to RF21-3 flies, the RF9-15 group showed a significant accumulation of

fructose-6-phosphate right after the time of feeding, indicating that fructose-6-phosphate was not metabolized to glucosamine-6-phosphate by glutamine-fructose-6-phosphate amidotransferase (GFAT) (Figs. 2d and 4a). This would explain the lower UDP-GlcNAc peak level in the RF9-15 group (at ZT 24) as compared to that in the RF21-3 group (at ZT12) (Fig. 4a; t-test $p = 0.038$). Our results therefore suggest that GFAT, the rate-limiting enzyme in the HBP[16–19], represents a key clock-controlled regulator that inhibits metabolic flow into the HBP during mistimed feeding.

**Circadian clock regulates GFAT at post-transcriptional level.**
As the key molecular components of circadian clock are all transcription factors[34,35], we first investigated whether *gfat* is transcriptionally regulated by the circadian clock and/or feeding-fasting cycle. We assayed the mRNA levels of *gfat1* and *gfat2*, two functionally equivalent paralogues, in body tissues of WT RF21-3, WT RF9-15 and *per^0* RF21-3 groups. *gfat1* mRNA did not exhibit daily rhythmicity in all three groups (Fig. 4b). On the other hand, we observed robust oscillation of *gfat2* mRNA levels with a peak right after the respective feeding window in all three groups (Fig. 4c). Interestingly, comparison of WT and *per^0* RF21-3 flies revealed very similar daily *gfat2* mRNA profiles, suggesting that feeding time, rather than the molecular clock, is the key determinant in regulating rhythmic *gfat2* mRNA expression (Fig. 4c). Since the expression of *gfat2* is 5 to 10-fold higher than that of *gfat1* (Supplementary Fig. 4b–d), we speculate that *gfat2* is the major paralogue that contributes to GFAT enzyme activity in fly bodies. We verified this by assaying overall GFAT enzymatic activity in heterozygous *gfat2^{18A-14}/+* flies[36]. Compared to WT flies, the overall GFAT activity is significantly reduced in *gfat2^{18A-14}/+* flies (Supplementary Fig. 4e), supporting the contribution of *gfat2* to GFAT activity in fly bodies.

Since GFAT phosphorylation has been shown to oscillate in mouse liver over a 24-h period[20], we hypothesized that the circadian clock may modulate fly GFAT enzymatic activity by post-transcriptional regulation. To test this hypothesis, we assayed GFAT activity in the same three groups of flies described above over a 24-h cycle. When WT flies were subjected to TRF at natural feeding time (RF21-3), we observed robust oscillation of GFAT enzymatic activity in whole body tissues that peaked immediately after the time of feeding (Fig. 4d). In comparison, GFAT activity in *per^0* flies fed in the same time window showed significantly lower GFAT enzyme activity with no apparent daily oscillation despite similar patterns of *gfat2* mRNA expression (Fig. 4c). This suggests a vital role for the circadian clock in driving daily rhythms of GFAT activity through post-transcriptional regulation. Finally, we observed that the time of

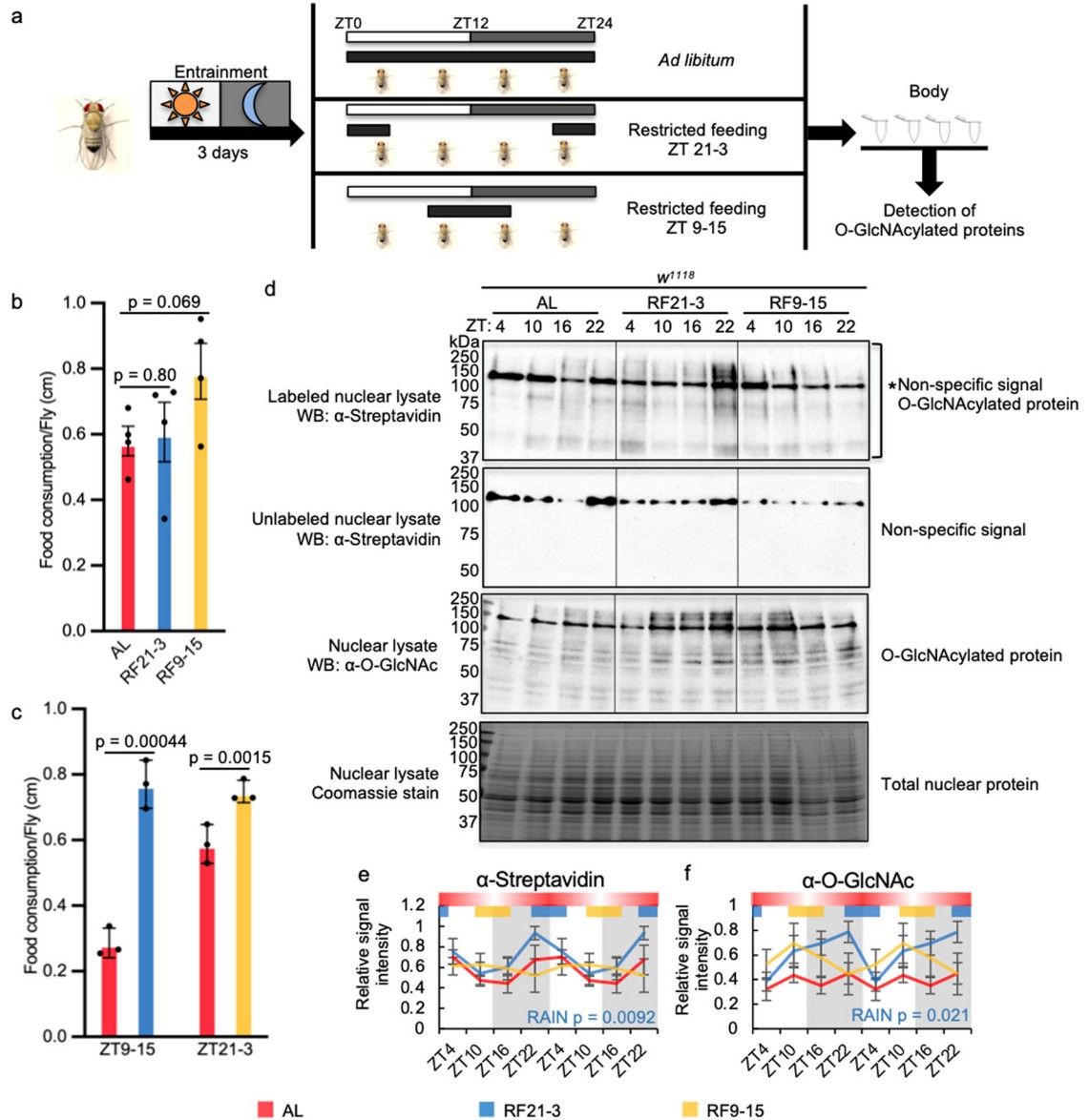

**Fig. 3 Time-restricted feeding (TRF) at natural feeding time strengthens the rhythm of protein O-GlcNAcylation. a** Schematic showing the study design. The black bars indicate the time when food was available. **b** Food consumption of AL, RF21-3 and RF9-15 groups over a 24-h period measured using CAFE assay ($n = 4$, in comparison to AL group, RF21-3 $p = 0.80$ and RF9-15 $p = 0.069$, two-tailed Student's $t$-test). **c** CAFE assay to compare food consumption of TRF groups to that of AL group during the indicated time window ($n = 3$, in comparison to AL group, RF21-3 $p = 0.0015$ and RF9-15 $p = 0.00044$, two-tailed Student's t-test). Data are presented as mean ± SEM. **d** Western blots showing daily rhythms in O-GlcNAcylation in flies fed AL, between ZT21-3, or ZT9-15. O-GlcNAcylation was detected using two methods: (top 2 panels) chemoenzymatic labeling in combination with immunoblotting with α-streptavidin and (third panel) immunoblotting with α-O-GlcNAc. Unlabeled samples (second panel) were processed in parallel to labeled samples (top panel) to identify non-specific signal. Total nuclear proteins stained by Coomassie blue (bottom) were used for normalization. The asterisk (top panel) indicates non-specific signal. Each biological replicate was performed independently with similar results ($n = 4$). **e, f** Quantification of protein O-GlcNAcylation detected using enzymatic labeling ($n = 4$; AL: $p = 0.045$, RF21-3: $p = 0.0092$, RF9-15: $p = 0.83$, RAIN; AL vs RF21-3: mesor $p = 0.05$, amplitude $p = 0.62$, phase $p = 0.3$, CircaCompare) and α-O-GlcNAc immunoblotting ($n = 4$; AL: $p = 0.83$, RF21-3: $p = 0.021$, RF9-15: $p = 0.69$; RAIN). Gray shading indicates the dark period of LD cycle. Colored bars at the top of graphs denote the feeding periods for TRF treatment groups, RF21-3 (blue) and RF9-15 (yellow). The red bars with gradient shading illustrate the representative feeding pattern for AL flies (data depicted from Fig. 2a), and the darker area indicates higher feeding activity. Data are double plotted, normalized (peak=1), and presented as mean ± SEM. The biological replicates of AL group that are quantified in (**e**) are the same four replicates quantified in Fig. 1b. In (**d**), a different biological replicate of AL group, other than the one shown in Fig. 1a, was ran together on the same gel with RF21-3 and RF9-15 groups.

feeding can also modulate GFAT enzymatic activity rhythm; WT flies subjected to TRF at unnatural feeding time (RF9-15) exhibited dampened GFAT activity rhythm that showed a phase difference of -3.48 as compared to that observed for the WT RF21-3 group (Fig. 4d). This could be attributed to the phase shift in *gfat2* mRNA rhythm that likely led to changes in daily

expression of GFAT protein. Taken together, our results indicate that nutrient-dependent *gfat2* mRNA induction and clock-controlled post-transcriptional mechanism collaborate to drive robust daily oscillation of GFAT activity.

To investigate the extent to which GFAT activity contributes to daily O-GlcNAcylation rhythm, we profiled daily O-

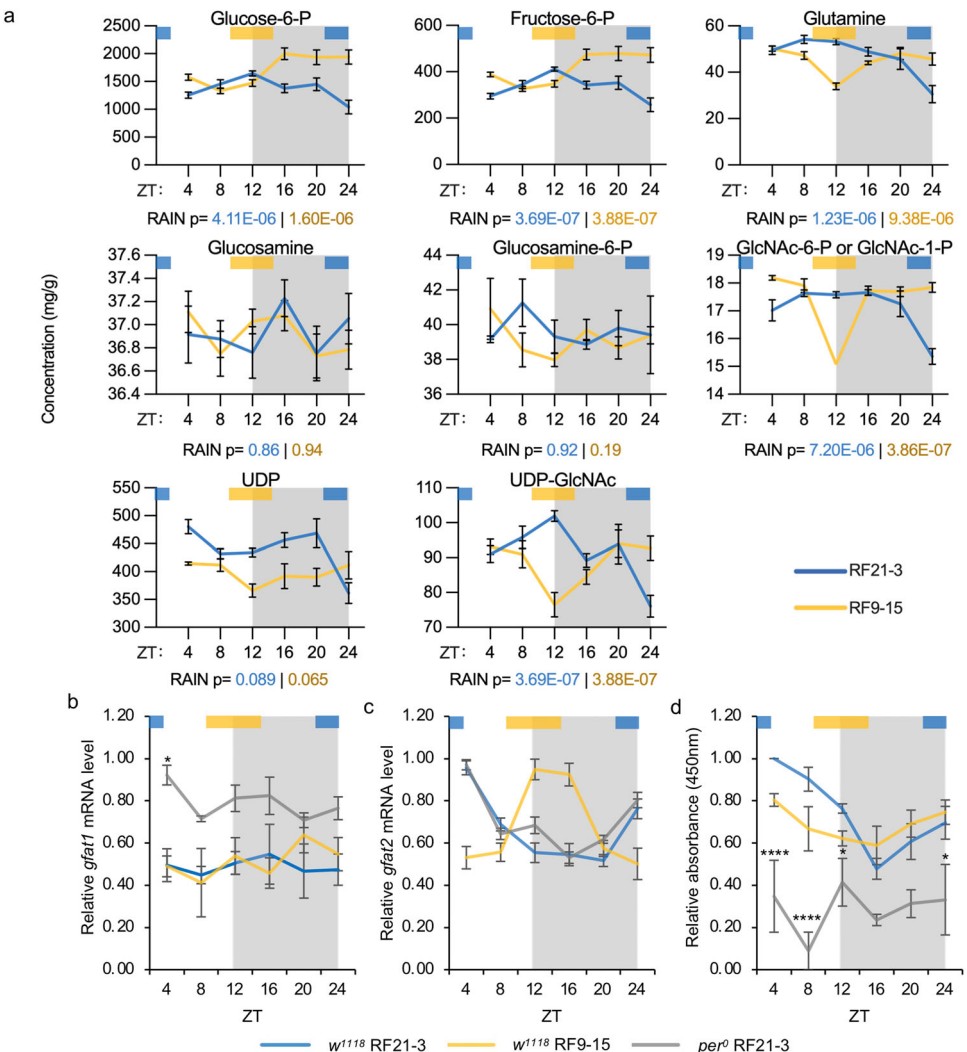

**Fig. 4 Glutamine-fructose-6-phosphate aminotransferase (GFAT) is a key enzyme that is regulated by circadian clock. a** Line graphs showing daily rhythms of HBP metabolites detected by targeted metabolomics in WT ($w^{1118}$) flies fed between ZT21-3 (blue) or ZT9-15 (yellow) ($n = 6$; p values from RAIN analysis are presented under each graph). Data were presented as mean ± SEM. **b**, **c** Expression of *gfat1* and *gfat2* mRNA in three groups of flies: WT RF21-3, $per^0$ RF21-3, and WT RF9-15. ($n = 3$; *gfat1* mRNA: WT RF21-3: $p = 0.73$, WT RF9-15: $p = 0.59$, $per^0$ RF21-3: $p = 0.76$; *gfat2* mRNA: WT RF21-3: $p = 1.58E{-}05$, WT RF9-15: $p = 0.00019$, $per^0$ RF21-3: $p = 3.89E{-}06$, RAIN); (*gfat2* mRNA: WT vs $per^0$ RF21-3: mesor $p = 0.23$, amplitude $p = 0.53$, phase $p = 0.66$; WT RF21-3 vs RF9-15: mesor $p = 0.95$, amplitude $p = 0.40$, phase $p = 1.12E{-}12$; CircaCompare). **d** Line graph representing daily GFAT enzymatic activity in WT ($w^{1118}$) flies fed between ZT21-3 or ZT9-15 and $per^0$ flies fed between ZT21-3 ($n = 3$; WT RF21-3: $p = 5.73E{-}07$, WT RF9-15: $p = 0.041$, $per^0$ RF21-3: $p = 0.92$, RAIN; WT RF21-3 vs RF9-15: mesor $p = 0.14$, amplitude $p = 0.01$, phase $p = 0.027$, CircaCompare). **b–d** Data are presented as mean ± SEM. Gray shading indicates the dark period of LD cycle. Colored bars at the top of graphs denote the feeding periods for TRF treatment groups, RF21-3 (blue) and RF9-15 (yellow). The asterisks denote the comparison between WT RF21-3 and $per^0$ RF21-3 using two-way ANOVA with *post-hoc* Tukey's HSD tests. *$p < 0.05$, ****$p < 0.0001$.

GlcNAcylation rhythms in WT and *gfat2*$^{18A-14}$/+ flies fed at natural feeding time (RF21-3). Using both O-GlcNAc chemoenzymatic labeling (Supplementary Fig. 4f, g) and western blotting by α-O-GlcNAc antibody (Supplementary Fig. 4 f and 4 h), we observed that rhythmicity of daily O-GlcNAcylation was attenuated in *gfat2*$^{18A-14}$/+ flies. These results indicate that GFAT indeed represents a key HBP enzyme that integrates metabolic and circadian signals to drive daily protein O-GlcNAcylation rhythm.

**Circadian and metabolic input regulate daily OGT and OGA rhythms.** In addition to GFAT, the two key enzymes that drive O-GlcNAc cycling, OGT and OGA[16–19], could also represent circadian effectors and contribute to the regulation of daily protein O-GlcNAcylation rhythm. Whereas OGA has been reported

to cycle at both the transcript and protein level in fly head and mouse liver, daily oscillation of OGT has only been observed at the transcript level in mouse liver (Supplementary Table 1). We therefore sought to examine the effects of the molecular clock and nutrient input on OGT and OGA expression to further improve our model describing key parameters that regulate daily O-GlcNAcylation rhythm.

Using nuclear extracts of WT AL, WT RF21-3, WT RF9-15 and $per^0$ RF21-3 flies, we found that both the molecular clock and nutrient input can regulate OGT protein expression (Fig. 5a, b, d, e). In WT AL flies, we observed that OGT displayed weak oscillation (rhythmicity was not significant based on RAIN) (Fig. 5a, b). Our results mirror observations reported in published studies using flies fed *ad lib*[23,24]. However, when WT flies were fed at natural feeding time (RF21-3), the daily oscillation of OGT

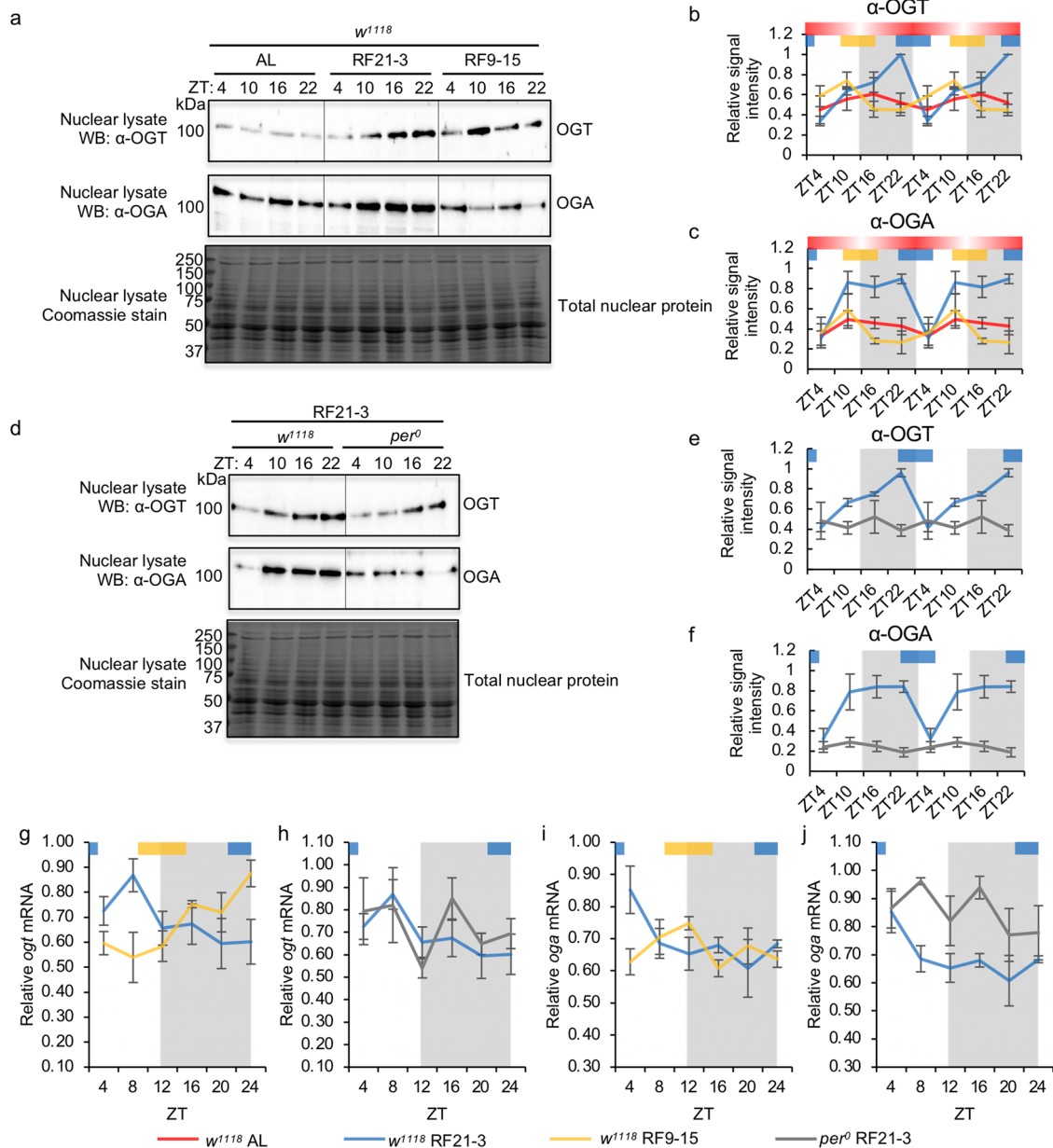

**Fig. 5 The molecular clock and the timing of food consumption regulate OGT and OGA protein levels. a** Western blots showing daily rhythms of OGT (top panel) and OGA protein (middle panel) in WT flies ($w^{1118}$) fed *ad libtum*, between ZT21-3 (natural feeding time), or between ZT9-15 (unnatural feeding time). Total nuclear proteins stained by Coomassie blue (bottom panel) were used for normalization. Each biological replicate was performed independently with similar results ($n = 4$). **b**, **c** Quantification of daily OGT protein rhythm ($n = 4$; AL: $p = 0.62$, RF21-3: $p = 3.92E-06$, RF9-15: $p = 0.036$, RAIN; WT RF21-3 vs RF9-15: mesor $p = 0.075$, amplitude $p = 0.25$, phase $p = 3.56E-06$, CircaCompare) and OGA protein rhythm ($n = 4$; AL: $p = 0.92$, RF21-3: $p = 0.051$, RF9-15: $p = 0.31$; RAIN). **d** Western blots showing daily rhythms in OGT (top panel) and OGA protein (middle panel) in WT ($w^{1118}$) and $per^0$ flies fed between ZT21-3. Each biological replicate was performed independently with similar results ($n = 4$). **e**, **f** Quantification of OGT ($n = 4$; WT: $p = 5.63E-07$, $per^0$: $p = 0.68$, RAIN) and OGA protein levels ($n = 4$; WT: $p = 0.038$, $per^0$: $p = 0.41$, RAIN). Data are double plotted. **g**, **h** Expression of *ogt* mRNA in three groups of flies: WT RF21-3, $per^0$ RF21-3, and WT RF9-15. ($n = 3$; WT RF21-3: $p = 0.041$, WT RF9-15: $p = 0.0033$, $per^0$ RF21-3: $p = 0.94$, RAIN; WT RF21-3 vs RF9-15: mesor $p = 0.84$, amplitude $p = 0.60$, phase $p = 2.36E-06$, CircaCompare). **i**, **j** Expression of *oga* mRNA in three groups of flies: WT RF21-3, $per^0$ RF21-3, and WT RF9-15. ($n = 3$; WT RF21-3: $p = 0.044$, WT RF9-15: $p = 0.22$, $per^0$ RF21-3: $p = 0.53$; RAIN). The biological replicates of WT RF21-3 that are quantified in panel (**h**) and (**j**) are the same three replicates quantified in panel (**g**) and (**i**) respectively. Gray shading indicates the dark period of LD cycle. Colored bars at the top of graphs denote the feeding periods for TRF treatment groups, RF21-3 (blue) and RF9-15 (yellow). The red bars with gradient shading illustrate the representative feeding pattern for AL flies (data depicted from Fig. 2a), and the darker area indicates higher feeding activity. Data are normalized (peak=1) and are presented as mean ± SEM.

expression was strongly enhanced with a peak around ZT 22 (Fig. 5a, b). In WT flies fed at unnatural feeding time (RF9-15), daily OGT protein rhythm was dampened and exhibited a phase shift that corresponded with the shift in feeding time (Fig. 5a, b).

These results suggest that timing of nutrient input can impact the amplitude and phase of OGT protein rhythm in the presence of an intact molecular clock. In $per^0$ RF21-3 flies, OGT protein level was arrhythmic and remained low throughout the day-night cycle

even when fed at natural feeding time (Fig. 5d, e), pointing to a vital role of an intact molecular clock in regulating daily OGT rhythm. Taken together, we conclude that the molecular clock is critical for maintenance of daily OGT oscillation, while the timing of nutrient input modulates the amplitude and phase of OGT rhythm.

As in the case of OGT, both the presence of an intact molecular clock and nutrient input at natural feeding time (RF21-3) are necessary to produce robust daily oscillation of OGA protein (Fig. 5a, c, d, f). We observed that nutrient input at natural feeding time (WT RF21-3) enhanced the low amplitude oscillation of OGA rhythm displayed by WT AL group (Fig. 5a, c). In WT RF9-15 group, OGA protein oscillation was severely dampened and statistically arrhythmic (Fig. 5a, c). Finally, we found that $per^0$ flies displayed arrhythmic OGA expression even when fed at natural feeding time (RF21-3) (Fig. 5d, f).

To determine if OGT and OGA protein rhythms are regulated by the molecular clock and nutrient input at the transcriptional level, we detected daily rhythms in $ogt$ and $oga$ transcripts in WT RF21-3, WT RF9-15, and $per^0$ RF21-3 flies. Comparing WT and $per^0$ flies fed between ZT21-3 will provide insights into the necessity of an intact molecular clock in shaping daily $ogt$ and $oga$ mRNA rhythms. On the other hand, comparing WT flies fed between ZT21-3 and ZT9-15 will reveal the sensitivity of $ogt$ and $oga$ mRNA rhythms to changes in timing of nutrient input in the presence of an intact clock. In WT flies fed between ZT21-3 (natural feeding time), we observed robust daily oscillations of $ogt$ and $oga$ transcripts, which were not observed in $per^0$ RF21-3 flies (Fig. 5h, j). These results indicate that the circadian clock regulates both $ogt$ and $oga$ at the transcriptional level. It is interesting to note that there is a noticeable lag period between the mRNA and protein peaks for both $ogt$ and $oga$, suggesting additional roles by post-transcriptional regulation to shape daily OGT and OGA protein rhythms. In the presence of an intact clock, we found that the timing of nutrient input can impact both $ogt$ and $oga$ mRNA rhythms. Whereas $ogt$ maintained robust daily rhythmicity with a phase shift that corresponded to the shift in feeding time when WT flies were fed between ZT9-15 (unnatural feeding time) (Fig. 5g), the mRNA rhythm of $oga$ was severely dampened (Fig. 5i). In sum, the molecular clock is a key regulator of daily OGT and OGA protein rhythms, and this regulation starts at the transcriptional level. Furthermore, nutrient input can enhance the amplitude and influence the phase of these rhythms, highlighting the importance of coordination between circadian and metabolic signals.

## $per^0$ flies fed at natural feeding time display impaired O-GlcNAcylation rhythm.

If clock-regulated rhythms of GFAT activity as well as OGT and OGA expression are important in generating daily protein O-GlcNAcylation rhythms, it is conceivable that arrhythmic clock mutant flies would exhibit impaired O-GlcNAcylation even if fed at natural feeding time. We detected nuclear protein O-GlcNAcylation in body tissues of WT and $per^0$ flies under RF21-3 condition using chemoenzymatic labeling and western blotting by α-O-GlcNAc antibody. In comparison to WT, nuclear protein O-GlcNAcylation of $per^0$ flies did not exhibit clear daily rhythmicity even though they were fed during natural feeding time (Fig. 6a–c). Since results from CAFE assay confirmed that WT and $per^0$ flies consumed similar amount of food under TRF treatment (Fig. 6d, t-test $p = 0.35$), we ruled out the possibility that the altered O-GlcNAcylation rhythm in $per^0$ flies was due to different amount of food consumption. Our results therefore support that clock-controlled mechanisms mediated by GFAT, OGT, and OGA represent important buffering mechanisms to inhibit excessive O-GlcNAcylation when animals feed at unnatural time.

## A mathematical model to describe and predict O-GlcNAcylation rhythm.

To assess the extent to which feeding time and clock-controlled mechanisms contribute to daily protein O-GlcNAcylation rhythm, we formulated a mathematical model by simulating the results of WT flies subjected to TRF at natural feeding time (ZT21-3) (see Table 1 for equations and variables). In WT flies, the circadian clock drives feeding-fasting cycles to provide timely nutrient influx, which transiently induces the expression of $gfat2$ and promotes GFAT enzymatic activity (Fig. 7a). In addition, there is(are) clock-controlled mechanism(s) at the post-transcriptional level to stimulate GFAT enzyme activity after food intake. GFAT, the rate-limiting enzyme of HBP, increases the HBP metabolic flow to produce the end product, UDP-GlcNAc. As the donor substrate of protein O-GlcNAcylation, UDP-GlcNAc feeds into the rhythmic activity of OGT and increases O-GlcNAcylation to peak level. Subsequently, OGA removes the O-GlcNAc groups on proteins to reset the O-GlcNAcylation status every day (Fig. 7a). OGT and OGA activities are estimated based on their protein levels (Eqs. 15 and 16). By testing the parameters for predicting OGT activity, we found OGT protein rhythm to be an accurate proxy for OGT activity and corresponds well with O-GlcNAcylation rhythm. However, in order to correctly predict O-GlcNAcylation rhythm, the contribution of OGA protein to OGA activity rhythm had to be adjusted to a negligible level. Our mathematical model therefore supports that OGA protein level is a poor indicator of OGA activity. This suggests that post-translational modification of OGA may play an important role in regulating daily OGA activity to shape O-GlcNAcylation rhythm.

To validate this model, we input the appropriate parameters for the WT RF9-15 group and predicted daily rhythms in GFAT activity, UDP-GlcNAc, OGT and OGA protein, and protein O-GlcNAcylation levels (Fig. 7b–f and Supplementary Fig. 5b, d). In agreement with our experimental results, the model predicted dampened GFAT activity rhythm with a similar phase for the WT RF9-15 group as compared to the WT RF21-3 group (Figs. 4d, 7b and Supplementary Fig. 5d). In addition, the model predicted lower amplitude of UDP-GlcNAc rhythm that peaks at the opposite phase, consistent with the fact that timing of food consumption significantly impacts metabolite levels (Figs. 4a, 7c and Supplementary Fig. 5d). Finally, the model predicted lower amplitude of OGT protein rhythm with a phase shift, dampened OGA protein rhythm, and dampened O-GlcNAcylation rhythm (Figs. 3e, f, 5b, c and Supplementary Fig. 5d).

We also validated the model by comparing the predictions to the experimental data obtained for $per^0$ RF21-3 flies (Supplementary Fig. 5c, e). Without a functional molecular clock, our model predicted that the daily rhythms of GFAT activity, UDP-GlcNAc, OGT protein, OGA protein, and protein O-GlcNAcylation levels would all be dampened, thus recapitulating our experimental data (Figs. 4d, 5e, f, 6b, c, 7b–f and Supplementary Fig. 5e). In summary, our mathematical model successfully predicted the phase and amplitude of daily GFAT activity rhythms and rhythms of UDP-GlcNAc, OGT, OGA, and protein O-GlcNAcylation levels in WT RF9-15 and $per^0$ RF21-3 groups. This suggests that our model included the major parameters that regulate daily protein O-GlcNAcylation rhythm.

## Discussion

In this study, we sought to investigate the mechanisms by which nutrient influx through clock-controlled feeding-fasting cycles regulate time-of-day-specific protein O-GlcNAcylation and cellular protein functions. Despite the important roles of O-GlcNAcylation in maintaining cellular homeostasis, most

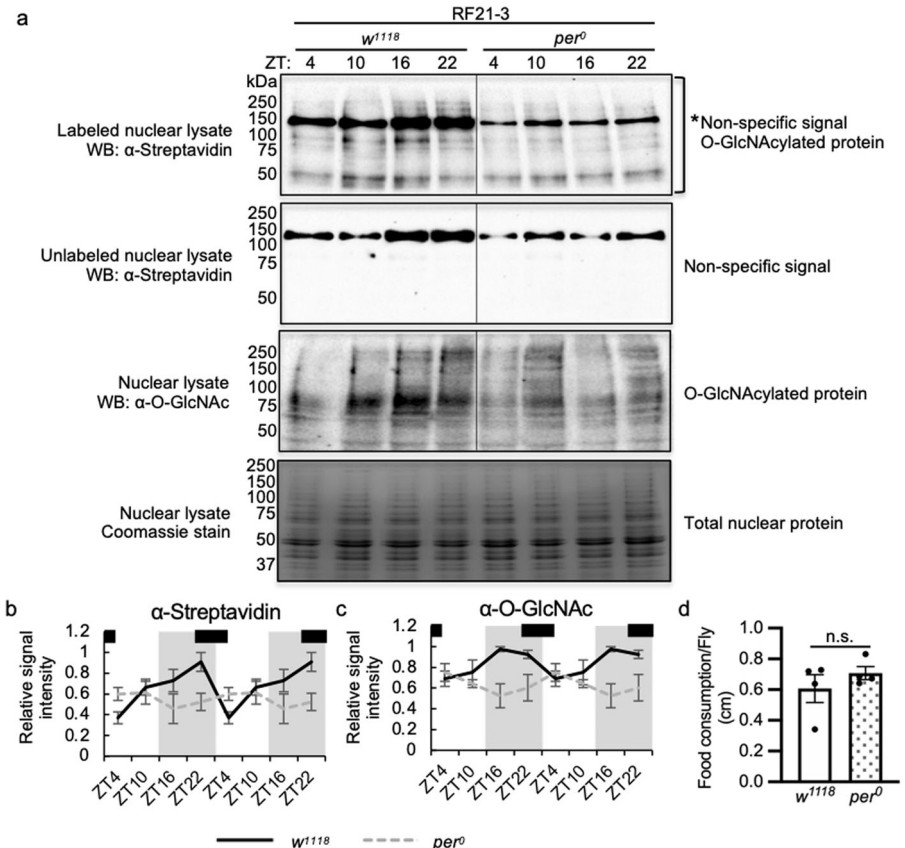

**Fig. 6 The circadian clock is necessary to generate daily rhythms in protein O-GlcNAcylation. a** Western blots showing daily rhythms in nuclear protein O-GlcNAcylation in WT ($w^{1118}$) and $per^0$ flies fed between ZT21-3 (natural feeding time). Protein O-GlcNAcylation was detected using two methods: (top 2 panels) chemoenzymatic labeling in combination with immunoblotting with α-streptavidin and (third panel) immunoblotting with α-O-GlcNAc. Unlabeled samples (second panel) were processed in parallel to labeled samples (top panel) to identify non-specific signal. Total nuclear proteins stained by Coomassie blue (bottom) were used for normalization. The asterisk (top panel) denotes non-specific signal. Each biological replicate was performed independently with similar results (n = 4). **b, c** Quantification of protein O-GlcNAcylation detected using enzymatic labeling (n = 4; WT: $p = 0.0096$, $per^0$: $p = 0.44$, RAIN; WT vs $per^0$: $p = 0.011$, DODR), and using α-O-GlcNAc (n = 4; WT: $p = 0.015$, $per^0$: $p = 0.28$, RAIN; WT vs $per^0$: $p = 0.017$, DODR). Data are double plotted. Gray shading indicates the dark period. Black bars on the top of graphs denote the feeding periods for flies, ZT21-3. **d** Food consumption of WT and $per^0$ flies over 24-hr period as measured by CAFE assay (n = 4; $p = 0.35$; two-tailed Student's t-test). Data are presented as mean ± SEM.

studies on protein O-GlcNAcylation were either conducted in tissue culture system or in metabolic disease models[18,19,37], and failed to take into account the intrinsic importance of O-GlcNAc cycling in the circadian timeframe. Here, we report daily rhythms of protein O-GlcNAcylation under normal physiological conditions by sampling WT fly tissues over a 24-hr period. Under *ad libitum* conditions, we established strong correlations between daily rhythms in food intake, UDP-GlcNAc, and cellular protein O-GlcNAcylation level. Despite the fact that O-GlcNAcylation is known to be nutrient sensitive, this is the first study to establish a clear relationship between feeding-fasting cycle and O-GlcNAcylation rhythm in whole animals. Furthermore, by subjecting flies to TRF, we uncovered clock-controlled buffering mechanisms that inhibits excessive O-GlcNAcylation when animals feed at non-optimal times in the day-night cycle. Specifically, we showed that the activity of GFAT and expression of OGT and OGA, which are regulated through integration of circadian and metabolic signals, play critical roles in mediating this buffering mechanism. The importance of these clock-controlled enzymes in shaping robust daily protein O-GlcNAcylation rhythms when animals feed at natural feeding time is also highlighted by the severely dampened O-GlcNAcylation rhythm of clock-defective $per^0$ mutant.

To describe key factors that regulate daily O-GlcNAcylation rhythm and identify potential parameters we have not yet considered, we generated a mathematical model that parameterized presence of an intact molecular clock and timing of food intake to predict protein O-GlcNAcylation rhythms (Table 1). Our model correctly predicted the dampened daily O-GlcNAcylation rhythms in WT RF9-15 and $per^0$ RF21-3 treatment groups (Fig. 7f), suggesting that the parameters in our model include major factors that impact O-GlcNAcylation rhythms. Nevertheless, our model may still lack some parameters that modulate protein O-GlcNAcylation rhythms. For example, OGT activity can be regulated by kinases, such as GSK3β, insulin receptor, AMPK and calcium/calmodulin-dependent protein kinase II (CaMKII)[38]. All of these kinases are sensitive to cellular metabolic status and are potentially regulated by either circadian clock or nutrient input. Additionally, since the parameters predicting OGA activity has to be adjusted to limit the effects of OGA protein on its activity, our mathematical model supports that OGA protein level is a poor indicator of its enzyme activity level. As both OGT and OGA are O-GlcNAcylated[39–41], daily O-GlcNAcylation rhythm may modulate the temporal activity of its own writer and eraser. Future investigations on mechanisms that regulate daily OGT and OGA activity will shed light on the

**Table 1 Mathematical model describing key parameters that regulate daily rhythms of protein O-GlcNAcylation.**

$G1_t = 0.60006 - 0.23683*C_t - 0.36690*C_{t-1}$ (1)

$G2_t = -0.15959 + 0.64048*F_{t-1} + 0.88571*F_{t-2}$ (2)

$G_t = 0.4 + a*3*G1_t + 2*G2_{t-2}$ (3)

$U1_t = 31.742 - 5.597*G_t - 4.992*G_{t-1}$ (4)

$U2_t = 32.915 - 34.274*F_t - 36.579*F_{t+1}$ (5)

$U_t = U1_{t-2} + U2_{t-1}$ (6)

$ogt_t = 0.06982 - a*(0.19496*F_t + 0.43217*F_{t-2} + 0.53953*F_{t-3})$ (7)

$OGT1_t = 0.73589 - 0.75087*ogt_{t+1} - 0.68392*ogt_t$ (8)

$OGT2_t = 1.06818 + 0.17796*C_t + 0.28562*C_{t-1}$ (9)

$OGT_t = 2.5*OGT1_t + a*OGT2_{t-1} - 1$ (10)

$oga1_t = 0.23416 - 0.24845*F_t + 0.29379*F_{t-1} + 0.18123*F_{t-2}$ (11)

$oga2_t = 1.3542 - 1.1107*C_t - 0.9980*C_{t-3}$ (12)

$oga_t = a*b*(oga1_{t-2}*oga2_t - 0.2)$ (13)

$OGA_t = 0.3379 - 4.3531*oga_{t+1} + 1.9031*oga_{t-1}$ (14)

$OGTAct_t = c*OGT_t + d$ (15)

$OGAAct_t = e*OGA_t + f$ (16)

$OXU_t = 0.4167*U_t*OGTAct_t/OGAAct_t$ (17)

$O_t = 0.2008750 - 0.0051100*OXU_t + 0.0071568*OXU_{t-2} + 0.0036060*OXU_{t-3}$ (18)

| Variables | Description |
|---|---|
| $G1_t$ | Clock-controlled post-transcriptional regulation on GFAT activity |
| $C_t$ | Clock-controlled mechanism(s) |
| $G2_t$ | Daily *gfat2* mRNA level |
| $F_t$ | Daily feeding pattern of flies |
| a | Parameter to indicate whether flies have an intact circadian clock. WT flies ($w^{1118}$): a = 1; clock mutant flies (*per⁰*): a = 0 |
| $G_t$ | Daily GFAT activity level |
| $U1_t$ | Level of UDP-GlcNAc calculated using GFAT activity |
| $U2_t$ | Level of UDP-GlcNAc calculated using feeding activity |
| $U_t$ | Overall daily UDP-GlcNAc level |
| $ogt_t$ | Daily *ogt* mRNA level |
| $OGT1_t$ | OGT protein level regulated by *ogt* mRNA |
| $OGT2_t$ | Translational regulation of OGT protein by clock |
| $OGT_t$ | Daily OGT protein level |
| $oga1_t$ | Feeding-regulated *ogt* mRNA level |
| $oga2_t$ | Clock-regulated *ogt* mRNA level |
| $oga_t$ | Daily *oga* mRNA level |
| b | Parameter to indicate whether flies eat at natural feeding time. RF21-3: b = 1; RF9-15: b = 0 |
| $OGA_t$ | Daily OGA protein level |
| $OGTAct_t$ | Daily OGT activity |
| c, d | Parameters to predict OGT activity from OGT protein level. In this model, c = 1 and d = 0 |
| $OGAAct_t$ | Daily OGA activity |
| e, f | Parameters to predict OGA activity from OGA protein level. In this model, e = 0.1 and f = 1 |
| $OXU_t$ | Interaction between daily UDP-GlcNAc level, OGT activity and OGA activity |
| $O_t$ | Daily O-GlcNAcylation level |

regulation of O-GlcNAcylation rhythm and further improve our mathematical model.

Future study to investigate how GFAT activity is regulated by clock-controlled post-transcriptional mechanism(s) is also warranted. Our results showed that GFAT activity is regulated by both feeding-induced gene expression and clock-controlled post-transcriptional mechanism(s). GFAT activity has been shown to be regulated by PKA- and AMPK-directed phosphorylation in vitro[42–46]. Interestingly, in mouse hepatic tissue, one of the previously identified PKA-directed phosphorylation sites on GFAT1, which activates its activity[42], was found to exhibit daily oscillation that peaks slightly after feeding phase[20]. This evidence supports our finding that GFAT activity peaks after natural feeding time in WT flies. We expect that future investigation on GFAT phosphorylation or other post-transcriptional mechanism will inform on the circadian regulation of GFAT activity.

Despite our findings that nuclear proteins extracted from fly body tissues exhibit pronounced daily rhythms in O-GlcNAcylation, the identity of cellular proteins that are regulated in this manner will need to be uncovered in circadian O-GlcNAc proteomic analysis. Nevertheless, given the diverse cellular proteins that are known to be regulated by O-GlcNAcylation, a significant portion of the proteome may be regulated in this manner. As explained earlier, this study focused primarily on O-GlcNAcylation analysis of total nuclear proteins. One of the major functions of nuclear proteins is transcription. Indeed, O-GlcNAcylation is known to modulate almost every step of transcription; the activities of transcriptional regulators including RNA polymerase II, histones, TATA-binding protein, and ten eleven translocation (TET) enzymes, are all heavily regulated by O-GlcNAcylation[18,19]. Moreover, O-GlcNAcylation is also known to be enriched in nuclear pore complexes, although the

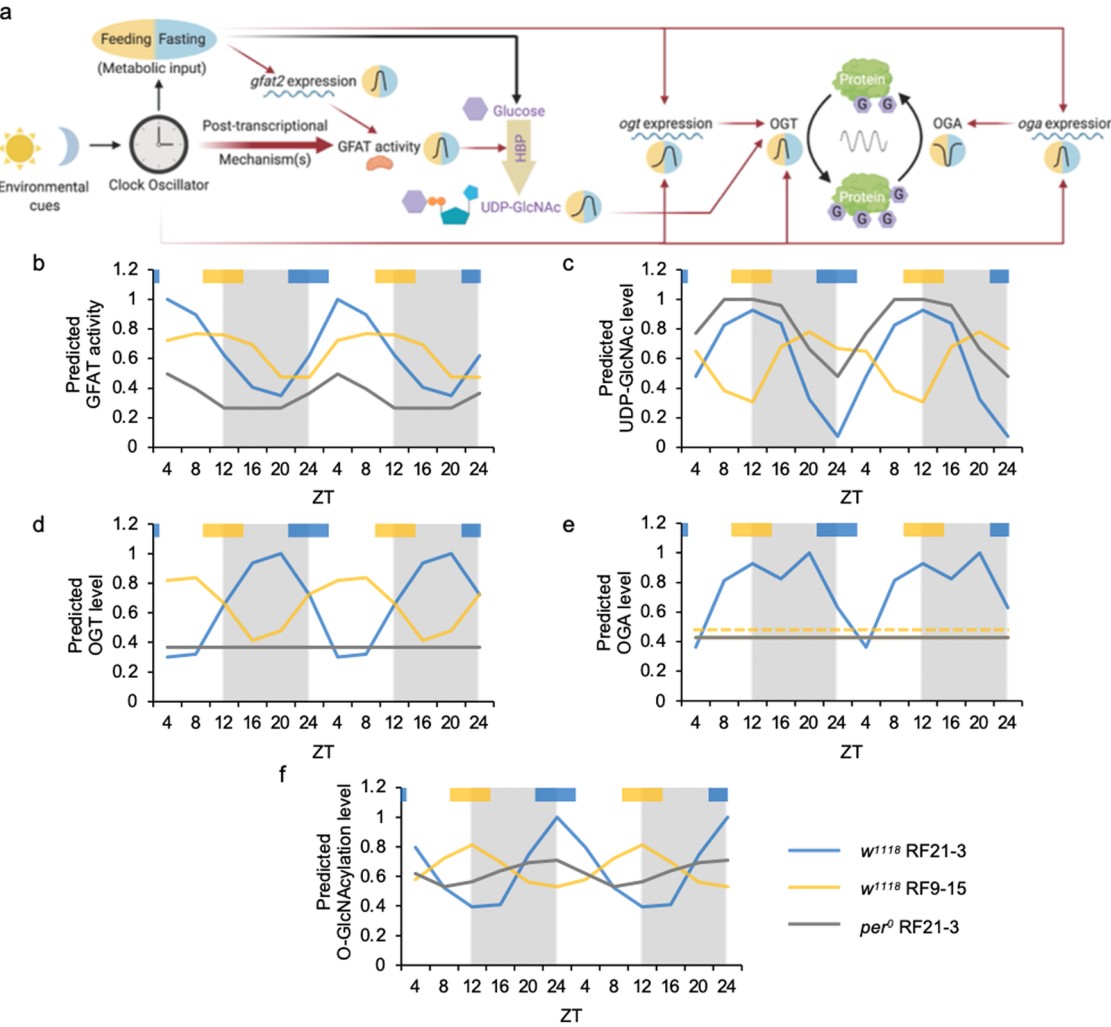

**Fig. 7 A mathematical model to describe and predict daily protein O-GlcNAcylation rhythm based on molecular clock function and timing of food intake. a** Schematic model showing the regulation of protein O-GlcNAcylation by the circadian clock through feeding activity and post-transcriptional mechanism(s). Yellow background indicates feeding period while the blue background indicates fasting period. Red arrows denote activation, and the thicker red arrows denote a stronger effect. **b–f** Line graphs showing daily rhythms of (**b**) GFAT activity, (**c**) UDP-GlcNAc, (**d**) OGT protein, (**e**) OGA protein and (**f**) protein O-GlcNAcylation levels as predicted using the mathematical model (Table 1). In (**e**), the predicted OGA protein levels are the same in WT RF9-15 and *per^0* RF21-3 flies, and the two curves overlap in the graph. Therefore, a dashed line is used to indicate OGA level in WT RF9-15 group. Time of feeding for WT RF21-3, WT RF9-15 and *per^0* RF21-3 fly groups were used as input for the model and the circadian clock is assumed to be functional in WT but not in *per^0* flies. Gray shading in graph area indicates the dark period of LD cycle. Colored bars at the top of graphs denote the feeding periods for TRF treatment groups, RF21-3 (blue) and RF9-15 (yellow).

specific function of O-GlcNAcylation in nucleocytoplasmic trafficking is still unclear[47]. O-GlcNAcylation is also essential for safeguarding the functions of multiple biological processes in the cytoplasm. Translational factors, ribosomal proteins, proteins in mitochondrial respiration chain, cytoskeletal proteins are all O-GlcNAcylated[19]. The prevalence of cellular protein O-GlcNAcylation suggests that it could represent a key mechanism by which organisms leveraged to reinforce the robustness of circadian physiology. We expect that future examination on cytoplasmic protein O-GlcNAcylation rhythm as well as circadian O-GlcNAc proteomics to identify substrate proteins would certainly reveal the physiological processes that are regulated in this manner.

Although most of this study focused on O-GlcNAcylation of proteins extracted from fly bodies as body tissues are known to be more metabolically sensitive in flies, it is interesting to note that a small portion of metabolites also oscillate in fly head tissues in our GC-MS data. Given that O-GlcNAcylation of PER and

CLOCK was shown to exhibit daily rhythms in fly heads[24,26], we expect that a significant number of fly brain proteins may also be sensitive to nutrient signals and could be regulated in a phase-specific manner by daily O-GlcNAcylation rhythm.

Since O-GlcNAcylation has been described in a wide range of species including *Caenorhabditis elegans*, fruit flies, mice, and humans[48,49], we expect that O-GlcNAcylation could represent a conserved molecular mechanism that integrates circadian and metabolic signals in animals. To investigate whether UDP-GlcNAc level also exhibits daily rhythmicity in the mouse model and could potentially drive rhythms in protein O-GlcNAcylation, we searched published circadian metabolomics datasets. We found that most of the studies were not able to detect UDP-GlcNAc, perhaps due to the use of different platforms and metabolomics methods[50–57]. Among the two studies that detected UDP-GlcNAc, one showed that TRF of clock mutant mice on high-fat diet (HFD) was sufficient to drive daily oscillation of UDP-GlcNAc in liver[58], while the other study failed to detect

rhythmic UDP-GlcNAc level in brown adipose tissues under *ad libitum* condition[59]. Based on these limited results, it is clear that at least some mammalian tissues would display daily rhythms in UDP-GlcNAc. Whether these rhythms in fact translate to rhythms in protein O-GlcNAcylation, as we observed in fly tissues, will need to be confirmed in future experiments.

In summary, we showed that HBP, OGT, and OGA integrate circadian and metabolic signals to drive daily rhythms in protein O-GlcNAcylation. Since we and others have previously reported that global manipulation of OGT and OGA activity and disruption of clock protein O-GlcNAcylation can alter circadian rhythms in whole animals[23–26], it is expected that daily O-GlcNAc cycling represents a key post-translational mechanism that regulates circadian physiology. Future investigations that examine functional consequences of protein O-GlcNAcylation and the interplay between O-GlcNAcylation and other PTMs will reveal how daily rhythms in O-GlcNAcylation manifest into rhythms in circadian physiology. Our results will facilitate future efforts to integrate key metabolic signaling pathways to obtain a systems-level understanding of metabolic regulation of circadian physiology. Finally, this study could shed light on the mechanisms by which lifestyle and eating habits common in modern society, especially night-shifted eating, contribute to the current epidemic of metabolic disorders and obesity as the misalignment of human lifestyles and natural day-night cycles continues to increase.

## Methods

**Animals**. $w^{1118}$; UAS-FLAG-*ogt* flies were described in Li et al.[26] *gfat2*[18A-14]/+ mutant flies[36] were obtained from Bloomington *Drosophila* Stock Center (BDSC stock no. 82445). *gfat2*[18A-14] mutant flies are homozygous lethal[36].

**Manipulation of metabolic input with time-restricted feeding**. Zero-to-four-day-old flies were entrained in 12 h light: 12 hr dark (LD) cycles at 25 °C for 3 days and split into three groups: *ad libitum* (AL), RF21-3, and RF9-15 groups. RF21-3 and RF9-15 groups were fed at ZT21-3 and ZT9-15, respectively, for 6 days. During the fasting period, flies were kept on 2% agar as water source. Flies were collected for O-GlcNAcylation analysis at the conclusion of the 6-day feeding treatments, i.e., flies are 10–14 day old when data were collected.

**CApillary FEeder (CAFE) assay to measure daily rhythms of fly feeding**. Daily feeding rhythms of flies fed *ad libitum* (AL) were measured by CAFE assay[4,31] after three days of entrainment in LD cycle. Food was provided to flies (groups of ten flies per biological replicate) as 5% sucrose solution in calibrated pipettes during a 24-h training period. Pipettes were replaced after the training period, and food consumption was measured at ZT 4, 10, 16, 22 for 2-h periods for two consecutive days. The evaporation rate was measured using pipettes without flies feeding on them. CAFE assay was also performed with TRF treatments, RF21-3 and RF 9-15. Just as for AL feeding, flies were fed with 5% sucrose solution in calibrated pipettes during the feeding period (ZT21-3 or ZT9-15) and changed into pipettes filled with water during the fasting period. One-day food consumption was measured at the conclusion of their feeding period.

**Chemoenzymatic labeling of O-GlcNAcylated proteins**. Nuclear proteins from fly bodies were extracted as previously described[26,60] with modifications. Fly bodies were ground in liquid nitrogen using mortar and pestles and resuspended in lysis buffer (20 mM HEPES pH7.5, 1 mM DTT, Roche EDTA-free protease inhibitor cocktail). After incubating for 15 min, a glass dounce homogenizer was used to disrupt the cell membrane. Samples were passed through cell strainers by centrifugation at $300 \times g$ for 1 min at 4 °C. Supernatant was collected and nuclei were pelleted by centrifuging at $4800 \times g$ for 15 min at 4 °C. The supernatant was collected as cytoplasmic fraction. The crude nuclear pellet was rinsed in wash solution (20 mM HEPES pH7.5, 10% glycerol, 150 mM NaCl, 0.1% TritonX-100, 1 mM DTT, 1 mM $MgCl_2$, 0.5 mM EDTA, 10 mM NaF, Roche protease inhibitor cocktail) twice, resuspended in nuclear extraction buffer (20 mM HEPES pH7.5, 10% glycerol, 350 mM NaCl, 0.1% TritonX-100, 1 mM DTT, 1 mM $MgCl_2$, 0.5 mM EDTA, 10 mM NaF, 100 $\mu$M PUGNAc, Roche protease inhibitor cocktail) and incubated for 30 min at 4°C on a rotator. After spinning at $16,000 \times g$ for 15 min at 4 °C, the supernatant was collected as nuclear fraction. 50 $\mu$g nuclear protein was precipitated using chloroform-methanol precipitation and dissolved in 40 $\mu$l 1% SDS with 20 mM HEPES pH7.9 by heating at 95 °C for 5-10 min. Samples were azide-labeled using Click-iT O-GlcNAc Enzymatic Labeling System and reacted with biotin alkyne using Click-IT Biotin Protein Analysis Detection Kit (Thermo Fisher

Scientific, Waltham, MA) following manufacturer's protocol. Unlabeled samples were processed in parallel for background deduction; the azide-labeling enzyme, GalT1 (Y289L), was excluded in these reactions. For PNGase F-treated samples, the nuclear proteins were first treated with PNGase F (New England Biolabs, Ipswich, MA) following manufacturer's protocol and then processed for chemoenzymatic labeling.

**Western blotting of protein extracts and protein quantification**. Western blot was performed after chemoenzymatic labeling to detect O-GlcNAcylated proteins using α-streptavidin-HRP (Cell Signaling Technologies, Danvers, MA) (1:3000). O-GlcNAcylated protein was normalized to total proteins stained with Coomassie blue (Bio-Rad, Hercules, CA) and the normalized signal intensity was presented as a proportion of the peak O-GlcNAcylation intensity (Peak=1). To confirm the results based on chemoenzymatic labeling, we also used α-O-GlcNAc (Abcam, Cambridge, MA) (1:1000) to detect O-GlcNAcylation in nuclear lysates. The O-GlcNAc antibody was validated using flies overexpressing OGT[26]. Our results indicated that there are higher levels of protein O-GlcNAcylation in the OGT overexpressor flies especially for proteins ranging from 37 to 250kD (Supplementary Fig. 1c). For this reason, only proteins within this size range were included in our quantification. Coomassie staining was performed to indicate equal amount of proteins in the labeling reactions and for normalization. The quality of nuclear and cytoplasmic extraction was validated by western blotting by α-Histone 3 (Sigma, St. Louis, MO) (1:2000) (Supplementary Fig. 1d). The α-OGT antibody (R5927, RRID: AB_2782954) (1:2000) was generated in-house (see next section) and validated using flies overexpressing FLAG-tagged OGT (Supplementary Fig. 1e). The α-OGA antibody (Proteintech, Rosemont, IL) (1:2000) was validated using *Drosophila* S2 cells expressing FLAG-tagged OGA (Supplementary Fig. 1f). FLAG-tagged recombinant proteins were detected using α-FLAG (Sigma, St. Louis, MO) (1:7000) to indicate the sizes of OGT and OGA proteins (Supplementary Fig. 1e, f). Secondary antibodies used for western blotting are as follows: α-mouse-HRP (1:1000) (GE Healthcare, Piscataway, NJ) for α-O-GlcNAc primary antibody, α-rabbit-HRP (1:2000) (Sigma, St. Louis, MO) for α-H3 primary antibody, α-mouse-HRP (1:5000) for α-FLAG primary antibody, α-rat-HRP (1:1000) (GE Healthcare) for α-OGT primary antibody, and α-rabbit-HRP (1:1000) for α-OGA primary antibody. Western blots data were imaged using Biorad Chemidoc and Image Lab software v6.1.

**Generating *Drosophila* OGT antibodies**. A fragment of *Drosophila* ogt cDNA (Flybase: FBgn0261403), including nucleotide number 1-1494, was subcloned into pET22-6xHis-NusA vector. The resulting plasmid, pET22-6xHis-NusA-*dogt* (1-1494), was transformed into BL21(DE3) competent cells. After large-scaled expression of OGT antigen, competent cells were lysed using extraction buffer (50 mM $Na_3PO_4$ pH8.0, 300 mM NaCl, 10% glycerol, 0.1% TritonX-100, 20 mM Imidazole, 5 mM β-mercaptoethanol). 6xHis-NusA-OGT was purified using IMAC Nickel Column (Bio-Rad) on NGC Chromatography System (Bio-Rad). The purified 6xHis-NusA-OGT was digested using TEV enzyme to separate OGT antigen and 6xHis-NusA tag. The OGT antigen and 6xHis–NusA tag mixture were applied to IMAC Nickel Column again to eliminate the 6xHis-NusA tag. For antibody production, OGT antigen was injected into rats by Covance Inc. (Princeton, NJ).

**Sample preparation for metabolomics analysis**. 10 mg of tissue per sample were ground using metal beads and extracted twice in 1 ml of methanol, chloroform and water (5:2:2) mixture for 20 min at 4 °C under constant agitation. After centrifugation at $14,000 \times g$ for 3 min, supernatants were pooled, evaporated to dryness under vacuum in a speedvac concentrator (Savant, ThermoFisher Scientific), and stored at −80 °C until GC-TOF MS or HILIC-TripleTOF MS/MS analysis.

**GC-TOF MS analysis**. Samples were analyzed using a GC-TOF MS approach as previously described[61]. Samples were randomized across the design using the MiniX database (http://minix.fiehnlab.ucdavis.edu/)[62]. A Gerstel MPS2 automatic liner exchange system (ALEX) was used to eliminate cross-contamination from sample matrix occurring between sample runs. 0.5 µl of sample was injected at 50 °C (ramped to 250 °C) in splitless mode with a 25 s splitless time. An Agilent 6890 gas chromatograph (Agilent, Santa Clara, CA) was used with a 30 m long, 0.25 mm i.d. Rtx5Sil-MS column with 0.25 µm 5% diphenyl film; an additional 10 m integrated guard column was used (Restek, State College, PA). Chromatography was performed at a constant flow of 1 ml/min, ramping the oven temperature from 50 °C for to 330 °C over 22 min. Mass spectrometry used a Leco Pegasus IV time of flight mass (TOF) spectrometer with 280 °C transfer line temperature, electron ionization at −70 V and an ion source temperature of 250 °C. Mass spectra were acquired from m/z 85–500 at 17 spectra/sec and 1750 V detector voltage.

Result files were exported to the servers and further processed by the metabolomics BinBase database[63]. All database entries in BinBase were matched against the Fiehn mass spectral library of 1200 authentic metabolite spectra using retention index and mass spectrum information or the NIST20 licensed commercial library (https://www.nist.gov/programs-projects/nist20-updates-nist-tandem-and-electron-ionization-spectral-libraries). Identified metabolites were reported if present in at least 50% of the samples per study design group (as defined in the MiniX database); output results were exported to the BinBase database and

filtered by multiple parameters to exclude noisy or inconsistent peaks. Quantification was reported as peak height using the unique ion as default. Missing values were replaced using the raw data netCDF files from the quantification ion traces at the target retention times, subtracting local background noise. Sample-wise metabolite intensities were normalized by the total signal for all annotated analytes. Daily quality controls and standard plasma obtained from National Institute of Standards and Technology (NIST) were used to monitor instrument performance during the data acquisition. Metabolite data were normalized to the sum of intensity peaks of NIST plasma metabolites to avoid the effect of machine drift among each sample. Four samples were determined as outliers by comparing the total peak intensity of individual replicates at each time point. Replicates that exhibited 10-fold differences were removed from the dataset (Supplementary Fig. 2b, c and Data 1). Data were then pareto scaled and heat maps were generated using Metaboanalyst 4.0[64]. The metabolomics dataset has been deposited in the open metabolomics database, Metabolomics Workbench (https://www.metabolomicsworkbench.org/), under accession no. [ST001110]. Unknown metabolites can be visualized and investigated using the BinBase identifier (Supplementary Data 1, column D) in https://binvestigate.fiehnlab.ucdavis.edu/#/.

**HILIC-TripleTOF MS/MS analysis.** 5 μl of resuspended sample was injected onto a Waters Acquity UPLC BEH Amide column (1.7 μm, 150 mm × 2.1 mm) with a Waters Acquity UPLC BEH Amide VanGuard pre-column (1.7 μm, 5 mm × 2.1 mm). Columns are maintained at 40 °C. The mobile phases were prepared with two solutions: (A) 10 mM ammonium formate and 0.125% formic acid (Sigma, St. Louis, MO) in ultrapure water and (B) 10 mM ammonium formate and 0.125% formic acid (Sigma, St. Louis, MO) in 95:5 v/v acetonitrile: ultrapure water. Samples were eluted at 0.4 mL/min with the following gradient: 100% (B) at 0–2 min, 70% (B) at 2–7 min, 40% (B) at 7.7–9 min, 30% (B) at 9.5–10.25 min, 100% (B) at 10.25–12.75 min, 100% (B) until 16.75 min. Mass spectra were collected using an AB Sciex TripleTOF 6600 (SCIEX, Framingham, MA) in ESI (−) mode. Electron ionization voltage was −4.5 kV and spectra were acquired from m/z 60–1200 at 2 spectra/sec. Internal standard mix were injected along with each sample to monitor instrument performance during the data acquisition. A standard HBP metabolite mix, with the amount of each metabolites ranging from 6.25 to 50 ng, was used to generate the standard curve.

Mass spectra were deconvoluted, aligned and identified using MS-DIAL v3.90[65]. The metabolite data were normalized to the sum peak intensity of internal standard mix. Standard curves of the HBP metabolites were generated by plotting the normalized peak intensity against the amount of compound injected on the column. The amount of each HBP metabolites in fly samples was then calculated and converted to concentration (mg/g tissue) based on the injection volume and the input amount of fly tissues. The targeted metabolomics dataset has been deposited to Metabolomics Workbench under the accession no. [ST001452].

**GFAT enzymatic activity assay.** GFAT enzymatic activity was measured as described in Srinivasan et al. (2007)[66] with the following modifications. Fly bodies were ground in extraction buffer (60 mM $KH_2PO_4$, pH7.8, 50 mM KCl, 1 mM EDTA, 1 mM DTT) and sonicated to disrupt the cell membrane. Cell lysate was collected after centrifugation at $16,000 \times g$ for 15 min at 4 °C. GFAT reactions were performed with 500 μg protein dissolved in reaction buffer (60 mM $KH_2PO_4$, pH7.8, 50 mM KCl, 1 mM EDTA, 1 mM DTT, 15 mM fructose-6-P, 15mM L-glutamine). Two reactions were set up for each sample: positive reaction and negative reaction. In positive reaction, the endogenous GFAT catalyzed the reaction between fructose-6-P and glutamine to produce glutamate, which was the readout of GFAT activity, and reactions were incubated at 25 °C for 90 min and terminated by heating up at 95 °C for 2 min. In negative reaction, all the proteins in reaction, including GFAT, were denatured right away by heating up at 95 °C for 2 min and therefore, the original level of glutamate in fly lysate is measured for background deduction. After centrifugation at $16,000 \times g$ for 15 min, supernatant was collected and filtered using 10 kD MWCO spin filters (Millipore Sigma, Burlington, MA) to deproteinate. The byproduct from GFAT catalyzed reaction, glutamate, was quantified using glutamate assay kit (Millipore Sigma) following the manufacturer's protocol. 18 μl of samples was used in each glutamate assay reaction. Absorbance was measured at 450 nm using a TriStar LD 941 microplate reader (Berthold Technologies, Oak Ridge, TN). To determine the activity of GFAT, the absorbance of negative reactions was subtracted from that of positive reactions.

**Analysis of gene expression.** Total RNA of fly body tissues was extracted using TRI reagent (Sigma, St. Louis, MO). Complementary DNA synthesis and real-time quantitative PCR were performed as previously described[26,60]. Quantitative PCR data were collected using Biorad CFX96 and CFX Maestro Software 4.1.2433.1219. Expression of target genes was normalized to non-cycling cbp20 (Supplementary Table 2).

**Statistical analysis.** RAIN (Rhythmic Analysis Incorporating Nonparametric methods)[67] was used to determine rhythmicity, amplitude, phase and period length of metabolites, protein O-GlcNAcylation, steady-state mRNA levels, and GFAT activity. Differences in daily rhythmicity were assessed using DODR (Detection of

Differential Rhythmicity)[68] and differences in parameters (mesor, amplitude and phase) between two rhythms were analyzed using CircaCompare (https://rwparsons.shinyapps.io/circacompare/)[69]. RAIN and DODR were performed in R v3.6.1. Significance in differences between treatments was determined using GraphPad Prism 8.0 (GraphPad Software, La Jolla California USA). The difference of gfat mRNA or GFAT activity levels was assessed using two-way ANOVA with post-hoc Tukey's HSD tests, while pair-wise comparisons between food consumption were analyzed using two-tailed Student's t-test with equal variance. The coefficients for cross-correlation between rhythms were calculated in R v3.6.1 using "astsa" package[70].

**Mathematical modeling.** We generated a mathematical model to describe and predict O-GlcNAcylation rhythms using timing of fly feeding activity and the presence of molecular clock function. The model is designed based on Fig. 7a. We simulated the data of WT fly feeding activity at natural time (ZT21-3) using "astsa" package in R v3.6.1[70], and calculated the rhythm of protein O-GlcNAcylation using the equations listed in Table 1. Description of variables is also included in Table 1. The input data for the model was the feeding activity of flies, which was first decomposed using "astsa" package to extract the rhythmic pattern (Supplementary Fig. 5a–c)[70]. The decomposed feeding data were used to calculate the levels of gfat2 mRNA. GFAT activity was determined based on both gfat2 mRNA and clock-controlled post-transcriptional mechanism(s). The clock-controlled post-transcriptional mechanism(s) was(were) simulated to peak at the same time of maximum GFAT activity in RF21-3 flies. Using the daily GFAT activity and feeding activity, the UDP-GlcNAc rhythm was calculated. We also modeled the regulation of OGT and OGA protein level by molecular clock functionality and timing of food consumption (nutrient input), and predicted OGT and OGA activities based on their protein levels. We calculated the interaction among OGT activity, OGA activity, and UDP-GlcNAc. Finally, O-GlcNAcylation rhythm was determined based on the interaction among OGT activity, OGA activity, and UDP-GlcNAc levels. Data were presented as a fraction of the highest level of GFAT activity, UDP-GlcNAc, OGT protein, OGA protein, or O-GlcNAcylation (Highest value = 1).

**Reporting summary.** Further information on research design is available in the Nature Research Reporting Summary linked to this article.

## Data availability

For GC-MS metabolomics, samples were randomized across the design using the MiniX database (http://minix.fiehnlab.ucdavis.edu/). GC-MS metabolomics database entries in BinBase database (https://binvestigate.fiehnlab.ucdavis.edu/#/) were matched against the Fiehn mass spectral library of 1200 authentic metabolite spectra using retention index and mass spectrum information or the NIST20 licensed commercial library (https://www.nist.gov/programs-projects/nist20-updates-nist-tandem-and-electron-ionization-spectral-libraries). The untargeted metabolomics dataset has been deposited in the open metabolomics database, Metabolomics Workbench (https://www.metabolomicsworkbench.org/), under accession no. [ST001110]. The targeted metabolomics dataset on HBP was deposited to Metabolomics Workbench under accession no. [ST001452]. All data generated or analyzed during this study are included in the published article and its supplementary information files. Source data for all figures are provided in Supplementary Data 4. Source data are provided with this paper.

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

## Acknowledgements

We would like to thank the West Coast Metabolomics Center at UC Davis for their technical support. pET22-6xHis-NusA vector is a kind gift from Dr. Carrie L. Partch. This work is supported by National Institutes of Health grants R01 GM102225 and R01 DK124068 to J.C.C., and U24 DK097154 to the West Coast Metabolomics Center. XL is supported by the China Scholarship Council Fellowship, the UC Davis Jastro Graduate Research Fellowship, Marv Kinsey Scholarship, and Sean & Anne Duffey and Hugh & Geraldine Dingle Research Fellowship.

## Author contributions

X.L., J.C.C. and O.F. designed research; X.L., I.B., A.C., T.P., C.A.T. and J.J. performed research; X.L., J.C.C., I.B., O.F., A.C., Y.H.L. and J.J. contributed to critical interpretation of the data. X.L. and J.C.C. wrote the paper.

## Competing interests

The authors declare no competing interests.
