## [Peer Review File · Nature Communications]

Reviewer comments, first round -

Reviewer #1 (Remarks to the Author):

I was asked to evaluate the mass spectrometry portion of this work. These experiments are extremely well designed, with all appropriate measures taken to ensure validity of quantitative measurements. The metabolite identification workflow is also solid. All relevant information is provided.

I did find one grammatical error: on at least 2 occasions, in lines 559 and 597, the authors used the word "grinded" rather than "ground."

Overall, very nice work.

Cheryl Lichti

Reviewer #2 (Remarks to the Author):

These authors and others have shown several key clock transcription factors in flies and mice exhibit daily O-GlcNAcylation rhythms that regulate their subcellular localization, activity, and stability. However, according to authors, it is not clear whether cellular proteins beyond the circadian oscillator also exhibit daily O-GlcNAcylation rhythms. In the current study, the authors showed that O-GlcNAcylation of total nuclear proteins oscillates over a 24-91 hr period in wild type (WT) flies. They have further demonstrated that O-GlcNAcylation rhythms showed a strong correlation with rhythms in food intake and HBP metabolites. Moreover, the authors demonstrated that manipulated timing of nutrient input using time-restricted feeding (TRF) to establish causal relationships between feeding time, HBP metabolites, and protein O-GlcNAcylation rhythm. Overall, this is an interesting follow-up manuscript from their previous finding and most of the assays are already established in the lab. Overall designing of the experiments is reasonable and most of the results are presented immaculately. The topic itself is quite interesting, and we need more study like this to the established role of feeding-fasting with circadian clock and circadian clock transcription factors. However, several issues need to address for the consideration to publish this in Nature Communications.

1. There is very little innovation linked with these findings as most of their finding is a follow-up from previous studies. However, some mechanistic detail is important, which could have been validated in vivo (see below).
2. One of the major limitations of this study is that the authors did not provide any genetic and physiological evidence and speculate several important which could have easily addressed in vivo? For example, the author speculates that *gfat2* is the major paralogue that contributes to GFAT enzyme activity in fly bodies. This is one of the examples, there are several places in the manuscript the authors should use genetic tools to address these important questions in vivo. Therefore, the author must provide genetic validation to support their biochemical and metabolic evidence for consideration to publish in Nature Communications.
3. Another major limitation of this study is not revealing the significance of their finding of metabolic diseases. The authors cited previous findings describing the roles of O-GlcNAcylation in maintaining cellular homeostasis and indicating that most studies on protein O-GlcNAcylation were either conducted in tissue culture system or metabolic disease models. However, the authors did not reveal how their findings can relate to the previous finding particularly about metabolic diseases.
4. The authors found the collection of proteins (Fig.1), ranging from 37 to 250 kD, are O-GlcNAcyated in the nuclei extracted from body tissues. Later they have indicated that some of the proteins band was non-specific however, did not mentioned detail how they determine specific vs non-specific bands. At the beginning of the results section, this is a misleading statement and need a detailed clarification.
5. It is hard to judge circadian rhythmicity from a 6-hour intervals data harvest. The

recommended time is every 2-3 hours intervals.

6. More specific clarification and justification are required on feeding rhythm conducted during feeding-fasting cycles of WT (both sex). The authors reported the two-days of data collection from three replicates. However, they did not provide how long they kept flies in feeding-fasting cycles before data collection. Generally, flies showed a lot of variability in their feeding behavior, when first kept in feeding-fasting cycles. Also, the age of the flies is important. Did the authors notice any relationship with feeding rhythm with sleep-wakeup cycles? If there any link to sleep disruptions with protein O-GlcNAcylation? If yes, how finding from this study fits with that link. It has been already established that feeding/fasting pattern mediated with TRF affects sleep. It is important to know about the finding of this manuscript.

7. Metabolomics analysis was done on the extract isolated from whole flies while some assays were conducted on fly body tissues. It is hard to predict if altered signaling from fat metabolism or something else. Therefore, the authors need to preciously need to address for motivation to take different tissue/ whole fly for each of the experiments and how these results will help address the overall outcome of this manuscript findings. The authors need to indicate about these limitations. Also, in the same section, the authors indicated that the exclusion of some data is questionable.

8. The authors indicated the RF9-15 group exhibited a severely dampened O-GlcNAcylation rhythm despite having access to food for the same duration of time and consuming roughly the same amount as the RF21-3 group. These are important but not surprising finding, however, the authors need to address whether this was only specific to dampened O-GlcNAcylation rhythm, under mistime feeding?

9. The following heading is misleading/incomplete and the following paragraph doesn't justify this "The circadian clock regulates daily oscillation of protein O-GlcNAcylation via multiple mechanisms"

Response Letter

We thank the reviewers for their constructive comments. As suggested, we performed a number of additional experiments necessary to support the claims we made in the manuscript. This includes generating and validating a new *Drosophila* OGT polyclonal antibody to detect daily OGT rhythm in various experiments. Below, we provide point-by-point responses that address the reviewers' concerns. Our responses are in black regular font, and the reviewers' comments are in *red italics font*. For the convenience of the reviewers, we are submitting a "marked-up" copy of the manuscript with key changes highlighted in **red font**.

Reviewer #1 (Remarks to the Author):

I was asked to evaluate the mass spectrometry portion of this work. These experiments are extremely well designed, with all appropriate measures taken to ensure validity of quantitative measurements. The metabolite identification workflow is also solid. All relevant information is provided.

I did find one grammatical error: on at least 2 occasions, in lines 559 and 597, the authors used the word "grinded" rather than "ground."

We apologize for the grammatical error. As suggested by the reviewer, we have corrected these errors (Lines #646, 705, 772).

Overall, very nice work.

Cheryl Lichti

Reviewer #2 (Remarks to the Author):

These authors and others have shown several key clock transcription factors in flies and mice exhibit daily O-GlcNAcylation rhythms that regulate their subcellular localization, activity, and stability. However, according to authors, it is not clear whether cellular proteins beyond the circadian oscillator also exhibit daily O-GlcNAcylation rhythms. In the current study, the authors showed that O-GlcNAcylation of total nuclear proteins oscillates over a 24-91 hr period in wild type (WT) flies. They have further demonstrated that O-GlcNAcylation rhythms showed a strong correlation with rhythms in food intake and HBP metabolites. Moreover, the authors demonstrated that manipulated timing of nutrient input using time-restricted feeding (TRF) to establish causal relationships between feeding time, HBP metabolites, and protein O-GlcNAcylation rhythm.

Overall, this is an interesting follow-up manuscript from their previous finding and most of the assays are already established in the lab. Overall designing of the experiments is reasonable and most of the results are presented immaculately. The topic itself is quite interesting, and we need more study like this to the established role of feeding-fasting with circadian clock and circadian clock transcription factors. However, several issues need to address for the consideration to publish this in Nature Communications.

1. There is very little innovation linked with these findings as most of their finding is a follow-up from previous studies. However, some mechanistic detail is important, which could have been validated in vivo (see below).

In this study, we expanded the investigation of daily O-GlcNAcylation rhythms beyond clock transcription factors. We identified daily rhythms in cellular protein O-GlcNAcylation as an important post-translational mechanism by which circadian and metabolic signals are integrated to maintain robust

circadian physiology. In the model we presented in this study, circadian physiology is regulated directly by modifying cellular proteins in a broad range of molecular pathways rather than as an indirect effect of modifying key clock proteins within the molecular oscillator. We respectfully disagree that there is little innovation.

Our results advanced the understanding of the maintenance of O-GlcNAc homeostasis by highlighting the intrinsic importance of O-GlcNAc cycling in the circadian timeframe. Over 1000 papers have established the important roles of O-GlcNAcylation in cellular homeostasis and pathogenesis of metabolic diseases, such as cardiovascular diseases, diabetes and cancer. This is the first study to investigate the diurnal modulation of the proteome by O-GlcNAcylation under normal physiological conditions in whole animals and more significantly establish the relationship between daily rhythms in protein O-GlcNAcylation to feeding-fasting cycles.

Finally, this study provides mechanistic insights into the benefits of time-restricted eating (TRE) and deleterious effects of mistimed eating common in modern society. Our results suggest that mistimed eating will affect rhythms in protein O-GlcNAcylation and its interplay with other PTMs, thereby disrupting circadian health.

*2. One of the major limitations of this study is that the authors did not provide any genetic and physiological evidence and speculate several important which could have easily addressed in vivo? For example, the author speculates that *gfat2* is the major paralogue that contributes to GFAT enzyme activity in fly bodies. This is one of the examples, there are several places in the manuscript the authors should use genetic tools to address these important questions in vivo. Therefore, the author must provide genetic validation to support their biochemical and metabolic evidence for consideration to publish in Nature Communications.*

Thank you for the reviewer's suggestion to improve our study. To verify that *gfat2* is the major paralogue that contributes to GFAT enzyme activity in fly bodies, we performed new experiments to detect overall GFAT activity in heterozygous *gfat2*^{18A-14/+} flies (*gfat2*^{18A-14} is a null mutant allele and is homozygous lethal) (Cotsworth, 2010). We observed that overall GFAT activity is significantly reduced in *gfat2*^{18A-14/+} flies when compared to WT flies (Supplementary Fig. 4e). In addition, we investigated whether *gfat2*^{18A-14/+} flies exhibit impaired O-GlcNAcylation rhythm while they were fed at natural feeding time (RF21-3) (Supplementary Fig. 4f-h). We observed that daily O-GlcNAcylation to be arrhythmic in these flies, thus highlighting the role of *gfat2* in regulating protein O-GlcNAcylation (Supplementary Fig. 4e-h). For description of the new results, see Lines #250-253 and 271-277.

3. Another major limitation of this study is not revealing the significance of their finding of metabolic diseases. The authors cited previous findings describing the roles of O-GlcNAcylation in maintaining cellular homeostasis and indicating that most studies on protein O-GlcNAcylation were either conducted in tissue culture system or metabolic disease models. However, the authors did not reveal how their findings can relate to the previous finding particularly about metabolic diseases.

We agree with the reviewer that it is interesting to study the causal relationship between abnormal daily O-GlcNAcylation rhythm and metabolic diseases. However, this is beyond the scope of this study. This study aims to investigate the regulation of O-GlcNAcylation under normal physiological conditions. We are indeed planning to study the link between daily O-GlcNAcylation rhythm and metabolic diseases in future studies, but we feel that it is important to first establish the relationship between daily feeding-fasting cycle and protein O-GlcNAcylation rhythm in normal conditions in whole animals.

4. The authors found the collection of proteins (Fig.1), ranging from 37 to 250 kD, are O-GlcNAcylated

in the nuclei extracted from body tissues. Later they have indicated that some of the proteins band was non-specific however, did not mentioned detail how they determine specific vs non-specific bands. At the beginning of the results section, this is a misleading statement and need a detailed clarification.

We apologize for the confusion. The method by which we determined specific vs non-specific bands is described in “Methods” section (Lines #661-663). We added the phrase “Please refer to the ‘Methods’ section for the use of unlabeled samples to identify non-specific signal” (Lines #124-125), where we first describe non-specific bands in the “Results” section.

5. It is hard to judge circadian rhythmicity from a 6-hour intervals data harvest. The recommended time is every 2-3 hours intervals.

We recognize the drawbacks of sampling with 6-hour intervals, but given the need for hundreds of flies per sample for protein O-GlcNAcylation labeling and metabolomics experiments, it is difficult to sample every 2-3 hours. We showed that our sampling frequency is sufficient to observe statistically significant rhythms in O-GlcNAcylation and metabolites. We feel that this is the most objective method to assess circadian rhythmicity. To ensure the rigor of our experiments, we performed >3 biological replicates for all our experiments. Additionally, we used multiple statistical tests to evaluate the rhythmicity of our measurements, including RAIN (Thaben and Westermarck, 2014), JTK_CYCLE (Hughes et al., 2010) and MetaCycle (Wu et al., 2016). The statistical significance of our data is consistent among these methods. For this reason, we chose to only show the RAIN p value in the manuscript. CircaCompare is also used to compare differences between daily rhythms (Parsons et al. 2020).

6. More specific clarification and justification are required on feeding rhythm conducted during feeding-fasting cycles of WT (both sex). The authors reported the two-days of data collection from three replicates. However, they did not provide how long they kept flies in feeding-fasting cycles before data collection. Generally, flies showed a lot of variability in their feeding behavior, when first kept in feeding-fasting cycles. Also, the age of the flies is important. Did the authors notice any relationship with feeding rhythm with sleep-wakeup cycles? If there any link to sleep disruptions with protein O-GlcNAcylation? If yes, how finding from this study fits with that link. It has been already established that feeding/fasting pattern mediated with TRF affects sleep. It is important to know about the finding of this manuscript.

As suggested by the reviewer, we clarified the information about the age of flies in “Methods” section (Line #630). Flies were 0-4 days when put into the feeding treatments. Flies were kept at feeding treatment (*ad libitum* or TRF) for 6 days before feeding assays or biochemical experiments were performed. Therefore, flies were 10-14 days old when data were collected.

We agree with the reviewers that the relationship between TRF, sleep and O-GlcNAcylation rhythm is an interesting area to pursue, but that is beyond the scope of this current study. There is previous evidence showing that TRF at natural feeding time improve the sleep duration in 5-week old flies (Gill et al. 2015: Fig 1e). In the future, it would be important to investigate the molecular mechanism by which TRF affect sleep and whether the effect of TRF on sleep is mediated at least partially by O-GlcNAcylation of key regulatory and neuronal proteins.

7. Metabolomics analysis was done on the extract isolated from whole flies while some assays were conducted on fly body tissues. It is hard to predict if altered signaling from fat metabolism or something else. Therefore, the authors need to preciously need to address for motivation to take different tissue/ whole fly for each of the experiments and how these results will help address the overall outcome of this manuscript findings. The authors need to indicate about these limitations. Also, in the same section, the authors indicated that the exclusion of some data is questionable.

We clarified the choice of tissues in the “Results” section (Lines #147-149, 171-173). The metabolomics experiments were first performed using heads and bodies separately. Our results show that metabolite levels, especially UDP-GlcNAc, oscillate more robustly in body tissues than in heads. We therefore concluded that body tissue is more metabolic sensitive than head tissue. Since we were investigating the effects of metabolites on O-GlcNAcylation, we expect to observe more drastic effects using body tissue than using head tissue. Therefore, we chose to use fly bodies in all subsequent experiments.

For the exclusion of metabolomics data, we followed Dr. Oliver Fiehn’s suggestion. Dr. Fiehn, the Director of West Coast Metabolomics Center at UC Davis, is an expert in metabolomics and has published extensively in the field of metabolomics. He suggested that it is common practice for metabolomic studies to include many replicates (we included 6 biological replicates in our study for this reason) and exclude ones that are obvious outliers. Outliers were not determined randomly but by comparing the total peak intensity of individual replicates at each time point. We excluded replicates that exhibited 10-fold differences comparing to the other replicates, since these unusual large differences are more likely due to technical errors instead of biological variations.

Dr. Cheryl Lichti, the expert reviewer who evaluated our metabolomic analysis, also confirmed that our analysis is appropriate and reasonable. She wrote, “I was asked to evaluate the mass spectrometry portion of this work. These experiments are extremely well designed, with all appropriate measures taken to ensure validity of quantitative measurements.”

8. The authors indicated the RF9-15 group exhibited a severely dampened O-GlcNAcylation rhythm despite having access to food for the same duration of time and consuming roughly the same amount as the RF21-3 group. These are important but not surprising finding, however, the authors need to address whether this was only specific to dampened O-GlcNAcylation rhythm, under mistime feeding?

As noted by the reviewer, RF9-15 group roughly consumed the same amount of food as the RF21-3 group. This is a control experiment to verify that the dampened O-GlcNAcylation rhythm in RF9-15 group was NOT due to any difference in food consumption between RF9-15 and RF21-3 groups. We performed this control experiment because our results clearly showed that the timing of feeding activity affects O-GlcNAcylation levels and rhythms. By comparing O-GlcNAcylation rhythm in RF9-15 and RF21-3 groups, we speculate that additional clock-controlled mechanism(s) other than timing of feeding activity are likely important in regulating daily O-GlcNAcylation rhythm. This led us to investigate the important roles of clock-controlled rhythms in GFAT activity as well as OGT/OGA expression in regulating O-GlcNAcylation rhythm in this study. All of these rhythms are also dampened in the RF9-15 group.

9. The following heading is misleading/incomplete and the following paragraph doesn’t justify this “The circadian clock regulates daily oscillation of protein O-GlcNAcylation via multiple mechanisms”

As suggested by the reviewer, we edited this title to better reflect the results of this Results subsection. We changed the title to “Feeding-fasting cycles regulate daily O-GlcNAcylation rhythm” (Line #175).

Reviewer 3:

The study by Liu et al. used flies as a model system to demonstrate that the total nuclear protein O-GlcNAcylation rhythms displayed strong correlation with the rhythms of food intake. They further showed that metabolic flux through the HBP, especially the enzymatic activity of the rate-limiting enzyme GFAT, may represent a key regulatory node for the rhythms of O-GlcNAcylation. Finally, the authors generated a mathematical model to describe the daily rhythms of O-GlcNAcylation. The logic of this study flows well, but the current study is rather preliminary. The study mainly focuses on the description of the

correlation between environmental cues and cellular O-GlcNAcylation. No molecular mechanisms were demonstrated to provide a real insight into the regulation of rhythms of O-GlcNAcylation. Regarding to the activity of GFAT, the study only described its correlation with O-GlcNAcylation. More experiments are needed to delve into the underlying mechanisms. In addition, given the current experimental results, the claim of the study is overstated.

Although we agree with the reviewer that our study leaves some questions unanswered, we regard some of those questions to be outside the scope of this current study. Nevertheless, to strengthen our manuscript and support the conclusions we stated, we have performed a number of additional experiments recommended by the reviewer to provide additional molecular insights into the regulation of daily O-GlcNAcylation rhythm. These specific experiments and corresponding results are described below in the point-point response.

To summarize, we believe our study is significant for the following reasons:

- This study advances our understanding of the intrinsic importance of O-GlcNAc cycling in the circadian timeframe. To date, over 1000 papers have established the important roles of O-GlcNAcylation in cellular homeostasis and pathogenesis of metabolic diseases, such as cardiovascular diseases, diabetes and cancer. Yet this is the first study to investigate the daily rhythms in protein O-GlcNAcylation under normal physiological conditions in whole animals AND establish the relationship between these rhythms and feeding-fasting cycles.
- This study identifies daily rhythms in protein O-GlcNAcylation as an important post-translational mechanism by which circadian and metabolic signals are integrated.
- This study provides mechanistic insights into the benefits of time-restricted eating (TRE) and deleterious effects of mistimed eating common in modern society. Our results suggest that mistimed eating will affect rhythms in protein O-GlcNAcylation and its interplay with other PTMs, thereby disrupting circadian health.
- As the regulation of cellular proteins by O-GlcNAcylation is now believed to be extensive and important in regulating a wide range of cellular processes and abnormal O-GlcNAcylation status has been reported as a hallmark of metabolic diseases and cancer, we believe our study will be of interest to the broad scientific community, beyond the field of circadian biology.

1) To eliminate the contamination from N-glycans, treatment with the enzyme PNGase should be included as an important control.

We agree with the reviewer that contamination from N-glycans is a potential problem for the labeling of O-GlcNAcylation proteins. As outlined in our results section (line #132-136), this was the reason we decided to focus on the analysis of nuclear proteins to avoid contamination from N-glycans in the endoplasmic reticulum, the Golgi apparatus, and on the cell surface. Nevertheless, as suggested by the reviewer, we included a control experiment in the revised manuscript where we compared the O-GlcNAcylation level of nuclear proteins with or without PNGase F treatment (Supplementary Fig. 1a-b). We observed that the O-GlcNAcylation level of the PNGase F-treated and untreated samples are NOT significantly different. We conclude that the O-GlcNAcylation signals observed using nuclear fraction have no detectable contamination from N-glycans as suggested by Spiro (2002).

2) OGT and OGA are key regulatory enzymes for the cellular recycling of O-GlcNAcylation. The authors

should investigate the rhythms of OGT/OGA expression, and further correlate them with the rhythms of protein O-GlcNAcylation.

We thank the reviewer for this suggestion, which helped us further improve our model describing the regulation of daily O-GlcNAcylation rhythm. As suggested by the reviewer, we performed a number of additional experiments and included them in the revision (detailed in Lines #279-327).

First, we investigated how the molecular clock and timing of food consumption (nutrient input) regulate OGT and OGA protein levels. We used four groups of flies: WT flies (w^{1118}) fed *ad libitum* (AL), WT flies fed at natural feeding time (RF21-3) or WT flies fed at unnatural feeding time (RF9-15), and clock mutant flies (per^0) fed at natural feeding time (RF21-3). We observed that an intact molecular clock is critical to generate robust daily OGT and OGA protein rhythms, and timing of nutrient input in the presence of an intact clock can influence the amplitude and/or phase of OGT and OGA protein rhythms (Fig. 5a-f). In order to assay OGT and OGA protein expression, we generated and validated a polyclonal antibody targeting *Drosophila* OGT (Lines #680-682, 692-702 and supplemental Fig. 1e). For detection of OGA, we used a commercial OGA antibody ordered from Proteintech, Inc. and validated it for detection of fly OGA (Lines #682-685, Supplemental Fig. 1f).

Second, we explored the mechanism by which the molecular clock and nutrient input regulate OGT and OGA proteins. We assayed *ogt* and *oga* mRNA level in three groups of flies: WT flies (w^{1118}) fed at natural feeding time (RF21-3) or unnatural feeding time (RF9-15), and clock mutant flies (per^0) fed at natural feeding time (RF21-3). *ogt* and *oga* mRNA levels generally correlate with their protein levels in all three groups, although mRNA rhythms peak earlier compared to protein rhythm (Fig. 5g-j). This suggests that the molecular clock and nutrient input can regulate OGT and OGA proteins through transcriptional mechanism(s) and the lag period between peak mRNA and protein rhythms is a result of post-transcriptional regulation, as in the case of many core clock genes.

Finally, given these new results, we added OGT and OGA protein levels as factors to predict protein O-GlcNAcylation rhythms in our mathematical model (Fig. 7a, d-f). We assessed OGT and OGA activity based on their protein levels (see revised Table 1 for equations). After testing a set of parameters for predicting OGT and OGA activity, we concluded that the rhythm of OGT protein could directly contribute to OGT activity and O-GlcNAcylation rhythm. However, OGA protein level is a poor indicator of OGA activity level. We therefore conclude based on our modeling approach that additional post-translational mechanism(s) could regulate OGA activity, which contributes to daily O-GlcNAcylation rhythm. Our findings highlight the utility of the modeling approach in formulating new hypothesis that can be explored in future studies. Please see lines #354-361 for more detailed description.

3) The authors only checked the global O-GlcNAcylation levels of nuclear proteins. Presumably, the key clock transcription factors are part of the nuclear proteins. How do their glycosylation levels change corresponding to the total glycosylation levels? This should serve as an important control for the experiment.

We have previously reported that *Drosophila* PERIOD (PER), a key clock transcription factor, exhibits daily rhythmicity in O-GlcNAcylation (Li et al. 2019). In this paper, we expanded the investigation of daily O-GlcNAcylation rhythms beyond clock transcription factors and identified daily rhythms in cellular protein O-GlcNAcylation as an important post-translational mechanism by which circadian and metabolic signals are integrated to maintain robust circadian physiology. In the model we presented in this study, circadian physiology is regulated directly by modifying cellular proteins in a broad range of molecular pathways rather than as an indirect effect of modifying key clock proteins within the molecular oscillator.

4) What are those proteins in the nuclear extract possessing rhythms of O-GlcNAcylation? The authors could use quantitative proteomics to identify those proteins. Even though the authors discuss this issue in the Discussion section, the review feels that it is important to include the data in the current study.

We agree with the reviewer that it is important to address the identity of O-GlcNAcylated proteins, but we feel that our study is a valuable contribution to the literature even without the quantitative proteomics experiments. The goal of this current paper is to investigate how daily O-GlcNAcylation rhythm is regulated. We certainly plan to identify O-GlcNAcylated proteins and molecular pathways in a future study. Characterization of protein O-GlcNAcylation and identification of O-GlcNAcylated proteins using quantitative proteomics are not trivial. We have initiated optimization experiments to achieve these goals in the mouse model where we can achieve tissue-specific resolution, which is not possible in flies given the size of the animal and the materials needed for these analyses.

5) Can the author modulate GFAT activity and investigate its impact on the rhythms of O-GlcNAcylation? For example, using GFAT specific inhibitor or siRNA? Is the GFAT transcription regulated by the key clock transcription factors? How is GFAT phosphorylation regulated in cells in response to food intake? These questions are important to shed light on the possible regulatory mechanism of the rhythms of O-GlcNAcylation.

As suggested by the reviewer, we utilized heterozygous *gfat2*^{18A-14/+} flies (*gfat2*^{18A-14} is a null mutant allele and is homozygous lethal) (Cotsworth, 2010) to investigate the role of GFAT in regulating O-GlcNAcylation rhythm. We confirmed that GFAT enzyme activity is reduced in *gfat2*^{18A-14/+} flies (Supplementary Fig. 4e) and observed that daily O-GlcNAcylation is arrhythmic in *gfat2*^{18A-14/+} flies (Supplementary Fig. 4f-h) (Lines #250-253 and 271-277). Our results showed that GFAT activity is important in regulating daily O-GlcNAcylation rhythm.

With regard to the regulation of GFAT transcription by the molecular clock, we detected *gfat1* and *gfat2* mRNA levels in WT flies and *per*⁰ flies (arrhythmic clock mutant) in the first version of our manuscript (Fig. 4b-c). Our results show that *gfat1* mRNA is not rhythmic in either WT or *per*⁰ flies, but its level is higher in *per*⁰ flies. We observed that *gfat2* mRNA is strongly regulated by food intake instead of the molecular clock (Lines #240-250). Based on these results, we concluded that although PER may influence the overall level of *gfat1*, *gfat1* and *gfat2* transcripts are not rhythmically regulated by key clock transcription factors.

Finally, with regard to the regulation of GFAT phosphorylation by food intake, previous papers have shown that GFAT is phosphorylated and regulated by PKA and AMPK. Both kinases are regulated by the nutrient status of cells or dietary conditions (e.g. PKA: O'Brien et al. 1998, Wilson and Roach, 2002, Budhwar et al. 2010; AMPK: Coughlan et al. 2015, Jeon et al. 2016). Therefore, it is likely that GFAT phosphorylation is regulated by nutrient signals in addition to being clock-regulated. We discuss these phosphorylation events in lines #420-423 to provide some examples of post-transcriptional regulation of GFAT activity that can potentially play a role in regulating daily O-GlcNAcylation rhythm. We agree with the reviewer that the contribution of GFAT phosphorylation is an important question to address in future efforts to understand the regulation of daily O-GlcNAcylation, as suggested by our results and mathematical model. We expect that this will require a stand-alone study, especially if we aim to tease apart GFAT phosphorylation that is regulated by nutrient signals vs the molecular clock.

Reviewer comments, second round -

Reviewer #2 (Remarks to the Author):

The authors did a good job in addressing my comments and questions/concerns raised by reviewer 3. With additional data along with supporting evidence, in my opinion, the finding of this manuscript is suitable for publications. I do not have any additional questions. Best wishes

Reviewer #3 (Remarks to the Author):

The authors have performed additional experiments to address most of my previous concerns. The manuscript is substantially improved. I am happy to recommend the publication in Nat Commun.

Response Letter

We thank the reviewers for their constructive comments throughout the review and revision process. Our responses are in black regular font, and the reviewers' comments are in *red italics font*.

Reviewer #2 (Remarks to the Author):

The authors did a good job in addressing my comments and questions/concerns raised by reviewer 3. With additional data along with supporting evidence, in my opinion, the finding of this manuscript is suitable for publications. I do not have any additional questions. Best wishes

We appreciate the reviewer's time in assessing the revision of our manuscript.

Reviewer #3 (Remarks to the Author):

The authors have performed additional experiments to address most of my previous concerns. The manuscript is substantially improved. I am happy to recommend the publication in Nat Commun.

We appreciate the reviewer's time in assessing the revision of our manuscript.